# Strain to Ground Motion Conversion of DAS Data for Earthquake Magnitude and Stress Drop Determination

5  Itzhak Lior[1], Anthony Sladen[1], E. Diego Mercerat[2], Jean-Paul Ampuero[1], Diane Rivet[1], Serge Sambolian[1]

[1]Université Côte d'Azur, CNRS, Observatoire de la Côte d'Azur, IRD, Géoazur, France
[2]CEREMA, équipe MouvGS, Sophia Antipolis, Valbonne, France.

10  *Correspondence to*: Itzhak Lior (itzhaklior22@email.com)

**Abstract.** The use of Distributed Acoustic Sensing (DAS) presents unique advantages for earthquake monitoring compared with standard seismic networks: spatially dense measurements adapted for harsh environments and designed for remote operation. However, the ability to determine earthquake source parameters using DAS is yet to be fully established. In particular, 15  resolving the magnitude and stress drop, is a fundamental objective for seismic monitoring and earthquake early warning. To apply existing methods for source parameter estimation to DAS signals, they must first be converted from strain to ground motions. This conversion can be achieved using the waves' apparent phase velocity, which varies for different seismic phases ranging from fast body-waves to slow surface- and scattered-waves. To facilitate this conversion 20  and improve its reliability, an algorithm for slowness determination is presented, based on the local slant-stack transform. This approach yields a unique slowness value at each time instance of a DAS time-series. The ability to convert strain-rate signals to ground accelerations is validated using simulated data and applied to several earthquakes recorded by dark fibers of three ocean-bottom telecommunication cables in the Mediterranean Sea. The conversion emphasizes fast body-waves 25  compared to slow scattered-waves and ambient noise, and is robust even in the presence of correlated noise and varying wave propagation directions. Good agreement is found between source parameters determined using converted DAS waveforms and on-land seismometers for both P- and S-wave records. The demonstrated ability to resolve source parameters using P-waves on horizontal ocean-bottom fibers is key for the implementation of DAS-based earthquake early 30  warning, which will significantly improve hazard mitigation capabilities for offshore earthquakes, including those capable of generating tsunami.

## 1 Introduction

In recent years, the implementation of distributed acoustic sensing (DAS) for seismological purposes is rapidly expanding, for both on land (e.g., Zhan, 2020; Fang et al., 2020; Jousset et al., 2018; Yu et al., 2019; Ajo-Franklin et al., 2019; Walter et al., 2020) and ocean-bottom (e.g., Sladen et al., 2019; Lior et al., 2021; Lindsey et al., 2019; Williams et al., 2019; Spica et al., 2020) applications. However, the use of DAS for several fundamental seismological tasks is yet to be fully established. These include earthquake location and source parameter (magnitude and stress drop) determination, both essential for harnessing DAS for earthquake early warning (EEW). But while application of source location have been investigated by recent studies (e.g., van den Ende and Ampuero, 2020; Lellouch et al., 2020; Lindsey et al., 2017), the ability to infer source parameters has not been investigated in detail (e.g., Lellouch et al., 2020).

One of the major hindrances for source parameter determination using DAS stems from the measurement type: DAS produces strain or strain-rate recordings while source models (e.g., Brune, 1970, Madariaga 1976; Sato and Hirasawa, 1973) rely on ground motions, i.e. displacements, velocities or accelerations. Thus, the ability to invert for the source properties using conventional methods depends on the ability to reliably convert strain (rate) measurements to ground motions. The conversion between strain (rate) to ground motion has been demonstrated by various previous studies (e.g. Daley et al., 2016; Lindsey et al., 2020; Paitz et al., 2020; Wang et al., 2018; Lior et al., 2021; van den Ende and Ampuero 2020). This conversion can be accurately done by spatial integration if one seismometer is co-located with the fiber, along straight portions that have uniform coupling (van den Ende and Ampuero 2020; Wang et al., 2018). When these conditions are unavailable, the common conversion approach consists of estimating the apparent phase velocities along the fiber and converting strain (rate) to ground velocity (acceleration) in the time- (e.g., Wang et al., 2018) or frequency- (e.g., Lindsey et al., 2020) domains, using the basic relations:

$$A = \dot{\epsilon}/p, \qquad\qquad (1a)$$

$$V = \epsilon/p, \qquad\qquad (1b)$$

where $\epsilon$, $\dot{\epsilon}$, $A$, $V$ and $p$ are the strain, strain-rate, acceleration, velocity and apparent phase slowness along the fiber, respectively. These equations are valid for a single plane wave.

Since different seismic phases travel at different velocities, and frequently in different directions, the apparent velocity required to convert strain (rate) to ground motions may rapidly vary, and a single time-invariant value may bias the analysis. In addition, velocities may vary along a fiber due to local velocity structure and fiber orientation variations. The ability to robustly convert

DAS records to ground motions for signals containing varying phase velocities is key for harnessing DAS for early-warning and hazard mitigation purposes. Determining phase velocities via frequency-wavenumber (FK) analysis (e.g., Lior et al., 2021; Lindsey et al., 2020; Paitz et al., 2020) can be challenging since sufficiently long cable segments and time-intervals are required to achieve adequate temporal and spatial frequency resolutions, in addition to delicate interpretation, as shown in later sections. To overcome this issue, a better suited approach to retrieve phase velocities as a function of time should be sought. In this study, we propose a method for continuous apparent phase velocity estimation using semblance-based local slant-stack transform (e.g., Neidell and Taner, 1971; Taner et al., 1979; Shi and Huo, 2019). This technique, commonly applied in exploration seismology (e.g., Tatham et al., 1983), is used here to estimate phase velocities as a function of time, facilitating a time-dependent conversion of DAS strain (rate) signals to ground motion records. We validate this conversion method using synthetic signals and apply it to ocean-bottom DAS earthquake records.

This manuscript is organized as follows. First, the slant stack algorithm is presented and validated using synthetic waveforms. Then, the approach is used to convert earthquakes recorded by ocean-bottom DAS to ground motions. Finally, source parameters are determined by fitting DAS observations with a source model, and compared with those determined using nearby seismometers. A comparison between the use of semblance-derived, and FK-derived apparent velocities is presented throughout the manuscript.

## 2 Slant stack transform for strain to ground motion conversion

Semblance-based local slant stack transform (e.g., Taner et al., 1979) is used to resolve apparent phase velocities as a function of time. This array-based technique measures the coherency (semblance) of plane waves recorded by several adjacent sensors. At each instance in time, a range of slowness values is tested to identify that with the maximum semblance. For each slowness, semblance is calculated by aligning the time-series recorded at different locations with respect to the middle station of a linear array (Shi and Huo, 2019):

$$sem(p_x, t) = \frac{1}{2L+1} \frac{\left[\sum_{j=-L}^{L} g\left(t+p_x(x_j-x_0)\right)\right]^2 + \left[\sum_{j=-L}^{L} h\left(t+p_x(x_j-x_0)\right)\right]^2}{\sum_{j=-L}^{L}\left[g\left(t+p_x(x_j-x_0)\right)^2 + h\left(t+p_x(x_j-x_0)\right)^2\right]}, \tag{2}$$

where $2L+1$ is the number of adjacent stations over which slowness is estimated, $x_j - x_0$ is the distance between station $j$ and the middle station, and $g(t)$ and $h(t)$ are the real and imaginary parts of the analytical signal associated with the seismic trace. The former is the

original trace and the latter is its Hilbert transform. The slowness with the highest semblance represents that of the most locally coherent plane wave at the specific time $t$. Including the Hilbert transform of the signal, $h(t)$, amounts to work with the analytical signal, i.e., $g(t) + ih(t)$. In fact, Eq. (2) results from applying the conventional definition of semblance to the analytical signal (Taner et al., 1979). This approach has the key advantage of allowing for reliable semblance calculation at the zero-crossings of the original signal, owing to the property that the amplitude of the analytical signal (which is the signal envelope) does not have zero-crossings.

For optimal slowness determination, fiber segment lengths should correspond to the longest wavelength of interest. However, when implementing the local slant-stack transform, segment lengths should be chosen to be upper bounded by the wavelength and the segment for which the signal remains coherent (coherency length), and lower bounded by the desired spatial and slowness resolutions (e.g., Ventosa et al., 2012). Spatial and slowness resolutions exhibit a trade-off since increasing the segment length will increase the slowness resolution and decrease the spatial resolution. Coherency lengths are expected to be small for DAS strain (rate) records since they may be dominated by scattered waves (e.g., Lior et al., 2021), are extremely sensitive to local media heterogeneities (e.g., van den Ende and Ampuero, 2020; Singh et al., 2019) and fiber coupling (e.g., Sladen et al., 2019; Lior et al., 2021). Thus, in the current application, short cable segments are used, as further described.

The process of converting DAS strain (rate) signals to ground velocity (acceleration) is detailed here, summarized in Fig. 1, and demonstrated using simulated data in the next section. Since seismic signals are a combination of various sources (e.g., earthquake waves, ambient noise, random signals), and may include dispersive waves, the signals need to be filtered at the frequency band of interest. Filtering will reduce noise and limit dispersive effect, however, simultaneous wave arrivals, complex propagation and dispersion effects are still expected. The slant stack transform is applied per virtual seismometer along the fiber, using the $2L + 1$ adjacent traces (L on each side). The range of examined slowness values is chosen to be between $-0.01 \text{ s m}^{-1}$ and $0.01 \text{ s m}^{-1}$ with $0.0002 \text{ s m}^{-1}$ slowness intervals (i.e., 100 slowness values), and at each time instance, the slowness value is determined based on the maximum semblance. The slowness time series (derived from semblance) may often be characterized by abrupt variations of value and sign (i.e., propagation direction) owing to complex wave propagation, interference and dispersion effects. Thus, a moving average, with window size set to be equal to the signal's longest period of interest, is applied to the absolute value of the slowness (preventing the averaging of

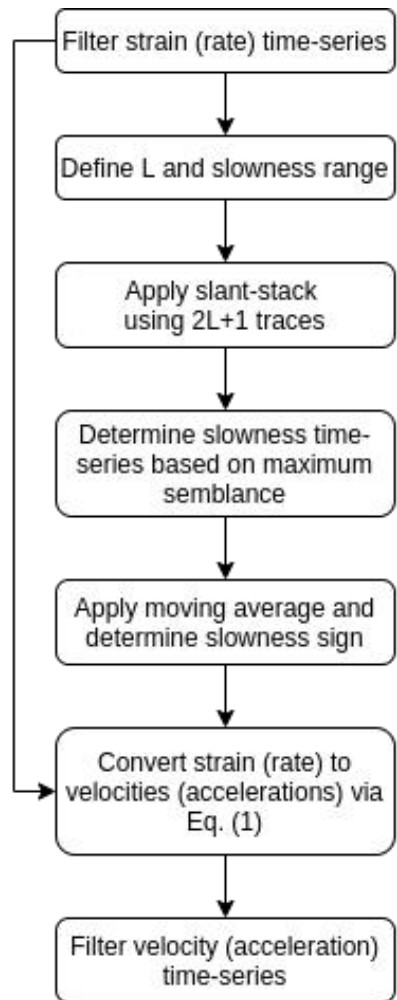

**Figure 1**: Flow chart detailing the conversion procedure from DAS strain (rate) signals to ground velocity (acceleration).

slowness values with different signs). The sign is then determined as the one that dominates each averaged window, i.e., the most recurrent sign. The filtered strain-rate signals are then converted to ground accelerations using Eq. (1). Slowness sign variations result in discontinuities in the converted strain-rate records, which are smoothed by additional filtering. The second filter may be chosen to be identical to the first, yet since it is applied to smooth discontinuities, different filters containing an upper frequency limit (i.e., low-pass or band-pass), may be applied. In following sections the same 4-pole zero-phase Butterworth filter is used for both filtering operations.

Throughout this manuscript, the semblance-based slowness determination method is compared to an FK-based method, applied as follows. Frequency-wavenumber transforms are calculated on the filtered strain (rate) signals in consecutive windows using the same number of adjacent traces used for semblance calculation. High amplitude temporal frequency ($f$) - spatial frequency ($\upsilon$) combinations are identified as those whose spectral amplitudes are higher than the
99th percentile of all amplitudes in the FK domain. The spectral amplitudes are summed separately for the two FK quadrant (positive $f$ and positive $\upsilon$ or positive $f$ and negative $\upsilon$) and slowness is estimated using data from the higher sum quadrant by fitting $f = \upsilon / p_x$. The slowness time-series is then smoothed and used for strain (rate) to ground motion conversion in the same manner as previously described for the semblance analysis.

## 3 Validation using simulated earthquake records

### 3.1 Simulated data

To validate the proposed method, simulated earthquake waveforms are produced for a simple 2D velocity model representing an underwater sedimentary basin. A basin was simulated to generate the commonly observed interactions of waves propagating in opposite directions. A
spectral-element based numerical simulation is done using the SPECFEM2D 7.0.0 code published under the CECILL V2 License (Komatitsch et al., 2012). The simple, though adequate, physical model is composed of two linear elastic sub-domains: a trapezoidal sedimentary layer with maximum depth of 100 m, characterized by P- and S-wave velocities of 1600 and 400 m s$^{-1}$, respectively, and density of 2000 kg m$^{-3}$ (yellow in Fig. 2a), overlying a bedrock with P- and S-wave
velocities of 2500 and 1200 m s$^{-1}$, respectively, and density of 2200 kg m$^{-3}$ (orange in Fig. 2a). These layers lie beneath a thin water layer at 20 m depth with P- and S-wave velocities of 1500 and 0 m s$^{-1}$, respectively, and density of 1000 kg m$^{-3}$ (green in Fig. 2a). The unstructured numerical mesh is generated using the Gmsh software (Geuzaine and Remacle, 2009) distributed under the terms of the GNU General Public License (GPL). It contains 2344 quadrilateral elements and, using
an interpolation polynomial order of 5, allows simulations up to 12 Hz maximum frequency. A double-couple point source is located at X = 2500 m, Z = 500 m and is time-modulated by a Ricker wavelet with a central frequency of 4 Hz, corresponding to a $M_w = 1$ earthquake. Receivers are regularly spaced at the bottom of the water layer from X = 400 m to X = 2600 m. Spatial and temporal sampling were set to be 5 m and 5 ms, respectively. These velocity waveforms represent
12.5 seconds of seismic recordings at 441 sensors, as seen in Fig. 2b. Since DAS can only record

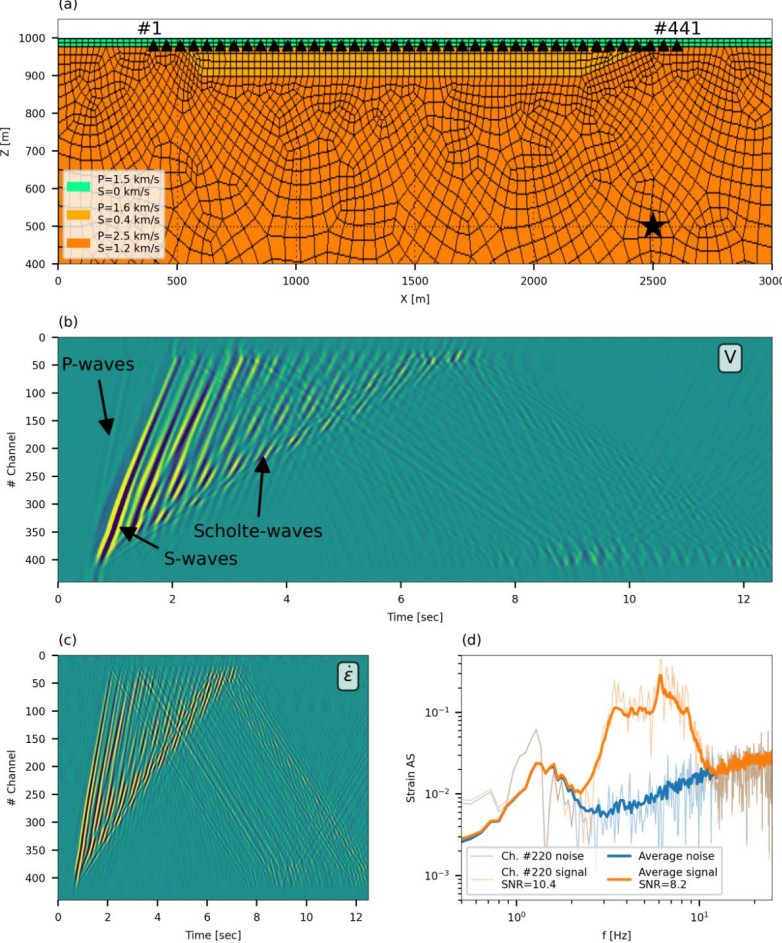

**Figure 2**: Velocity model and simulated velocity, strain-rate and noise data. Panel (a) shows the unstructured 2D numerical mesh: the double-couple source location is indicated by a black star and sensors are indicated by black triangles, equally spaced between X=400m to X=2600m (first and last sensor channel numbers are indicated). Panel (b) shows simulated velocity waveforms, with visible P-waves, S-waves and Scholte-waves. Panel (c) shows strain-rate signals added with ambient noise. Panel (d) shows simulated strain-rate and noise signals. Noise curves for channel #220 and averaged for all channels are indicated by semi-transparent and solid blue curves, respectively. Signal added with noise for channel #220 and averaged for all channels are indicated by semi-transparent and solid orange curves, respectively. SNR values are reported in the legend.

deformations along the optical fiber, simulated seismograms are single component, oriented along the array axis. These waves exhibit complex propagation patterns, with visible P- and S-waves, as

well as surface waves reflected from the edges of the basin, providing an excellent test-case for the proposed algorithm.

Simulated velocity waveforms were differentiated in time and space to obtain ground accelerations and strain-rates, respectively, noise was added to the later, and the ability to convert the latter to the former via the proposed slant-stack approach is examined. Strain-rate records were calculated as $\dot{e}(x) = \frac{V(x+GL/2)-V(x-GL/2)}{GL}$ and ground acceleration records were calculates as $A(t) = \frac{V(t+dt)-V(t-dt)}{2dt}$, where $V, GL$ and $dt$ are the simulated velocity, gauge length and temporal

sampling, respectively. Simulated strain-rate signals thus are characterized by gauge length and spatial sampling of 10 m and 5 m, respectively. To reliably generate ocean-bottom DAS records, ambient noise measurements were added to simulated strain-rates, keeping the noise records' spatial correlation (Fig. 2c). In the simulated water depth (i.e., 20 m), ocean-bottom DAS records are typically dominated by surface gravity waves (e.g., Lior et al., 2021), which may be easily

filtered due to their lower frequency content compared to the simulated earthquake. Thus, ambient noise recorded at a water depth of ~800 meters was used. These records are composed of instrumental noise and secondary microseisms in similar frequencies to those of the simulated earthquake (Fig. 2d). The added noise measurements were recorded on 22 July 2019 by an underwater cable deployed offshore Toulon, South of France (Sect. 4). Noise records, sampled at

100 Hz and 10 m (spatial sampling is equal to the gauge length), were resampled to match the simulated data using a 2D interpolation function. Noise records were then differentiated to strain-rate and rescaled to simulate challenging noise conditions, with an average signal-to-noise ratio (SNR) of 8.2 (Fig. 2d). Here, SNR was calculated as the root-mean-squares (RMS) ratio of average signal and noise amplitude spectra between 2 and 12 Hz. In-spite of the added noise, accelerations

converted from strain-rates are compared to simulated accelerations (derived from simulated velocities by finite-difference time-derivative). Noise was not added to the latter, constituting a stringent algorithm validation.

### 3.2 Strain-rate to ground acceleration conversion

       For the simulated data, waveforms were lowpass filtered at 12 Hz and short array segments

of 100 m (L=10) were used to calculate the semblance and to convert each strain-rate signal into an acceleration seismogram. These segments are 25% shorter than the longest apparent wavelength observed for P-waves and provide a sufficient compromise between slowness and spatial resolution, allowing for reliable strain (rate) to ground motion conversion, as further shown. FK-based analysis analysis was conducted on consecutive 1 second windows. An example of the

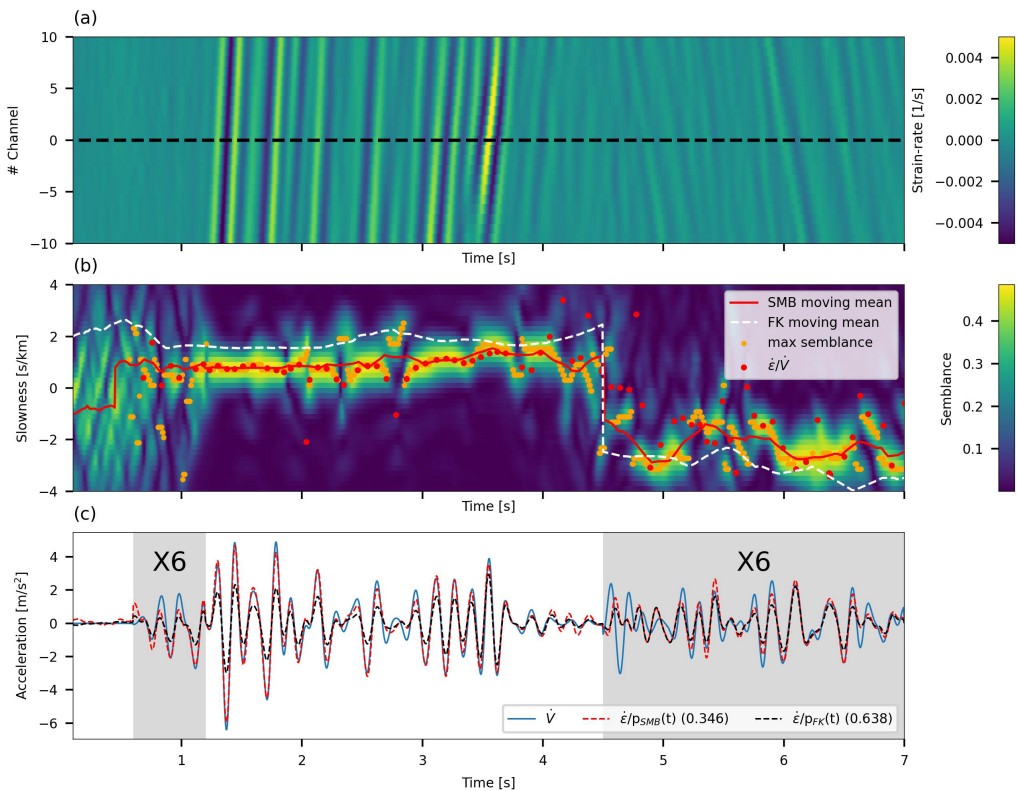

**Figure 3:** Slant-stack conversion for channel #220 of the simulated data. Top: strain-rate time series for 21 adjacent traces centered around channel #220 (dashed black line). Middle: Slowness as a function of time color coded by semblance values for channel #220. Slowness corresponding to the maximum semblance, and smoothed slowness are plotted in orange circles and a red curve, respectively. Smoothed slowness derived from FK-based analysis is indicated with a dashed white curve. Red dots correspond to the ratio between strain-rates and accelerations, both differentiated from the simulated velocity waveforms. Bottom: accelerations, time-differentiated from simulated velocities (blue curve), compared with accelerations, space-differentiated from simulated velocities and converted using semblance derived slowness (red curve) and FK derived slowness (black curve). Standard deviations of the residuals between converted strain-rates (red and black curves) and time-differentiated velocities (blue curve) are indicated in the panel legend. The signals in the gray regions have been amplified by a factor of 6 for easy comparison of low amplitude signals.

slant-stack conversion, applied to trace #220 of the simulated waveforms (Fig. 2b), is shown in Fig. 3. Panel (a) shows 7 seconds of strain-rate signals composed of P-waves (0.6-1.2 seconds), S-waves

(1.2-2.5 seconds) and surface waves (2.5-7 seconds), including reflected waves with opposite propagation direction (4.5-7 seconds). The semblance analysis in panel (b) shows apparent phase slowness, corresponding to the maximum semblance, and smoothed slowness in orange circles and red curve, respectively. Red dots correspond to the ratio between strain-rates and accelerations (both differentiated from the simulated velocity waveforms) plotted at acceleration maxima and minima, i.e., the optimal slowness values for strain-rate conversion. The similarity between the smoothed slowness (red curve) and optimal slowness for conversion (red dots) suggests that the resolved slowness may be reliably used for strain-rate to acceleration conversion. As expected, semblance and FK analysis are able to resolve different velocities for different phases. FK-derived velocities are significantly slower than semblance-derived velocities: semblance analysis resolves average apparent phase velocities of 1.3, 1.2 and 0.6 km s$^{-1}$ for P-waves, S-waves and surface waves, respectively, while FK analysis resolves average apparent phase velocities of 0.6, 0.6 and 0.4 km s$^{-1}$ for P-waves, S-waves and surface waves, respectively (in the previously indicated intervals). When no noise is added, the same velocities are determined, with the exception of a higher P-wave apparent velocity of 2.3 km s$^{-1}$ for the semblance-based method. The bottom panel compares the acceleration signal (blue curve, differentiated from simulated velocity) with strain rate converted accelerations using semblance-derived slowness (red curve) and FK-derived slowness (black curve).

Excellent agreement is observed between acceleration (blue curve in Fig. 3c) and converted strain-rate, when the latter is obtained using semblance-based slowness (red curve in Fig. 3c). This agreement persists for both fast (P-waves and S-waves) and slow (surface waves) phases, demonstrating the ability to reliably resolve phase velocities even for short duration waves and wavelengths longer than the fiber segments used for slant-stack analysis, e.g., P-waves. The low apparent velocities derived from FK analysis result in lower converted strain-rate signals (black curve in Fig. 3c) compared with acceleration records. Since reliable FK slowness estimation requires sufficiently long data segments, this method does not provide sufficient slowness resolution to resolve high apparent velocities in short spatial and temporal intervals. When different waves interfere, the single plane wave assumption in Eq. (1) does not hold, the phases of velocity and strain-rate signals may be misaligned (4.5-5.1 seconds) and phase velocities may not be reliably resolved, resulting in lower quality conversion. When the intensity of such effects fade, excellent agreement is again observed (5.1-7 seconds).

The use of time-dependent slowness is found to be particularly advantageous when velocities abruptly vary and when propagation direction changes. Specifically, the semblance

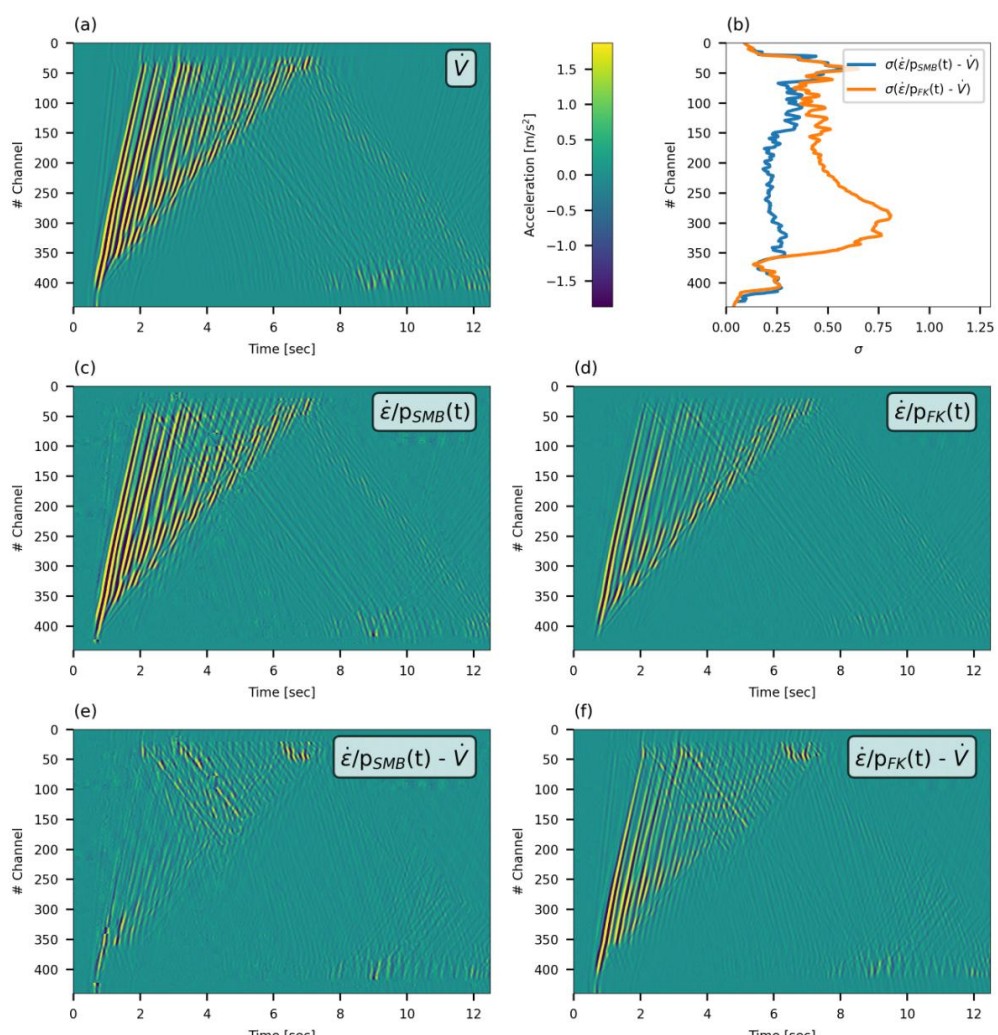

**Figure 4:** Comparison between simulated velocities time-differentiated to accelerations (panel a), and simulated velocities space-differentiated to strain-rate and converted to ground accelerations: using time-dependent slowness (panel c), and constant slowness (panel d). Panel (e) shows the residuals between accelerations in panels (c) and (a), and panel (f) shows the residuals between accelerations in panels (d) and (a). Standard deviations of the residuals for each recorded channel are plotted in panel (b) for the residuals in panel (e) (blue curve) and in panel (f) (orange curve).

approach is substantially more robust compared with the FK scheme when implemented on short spatial and temporal intervals. This is shown in Fig. 3c by comparing the FK-based converted

strain-rate (black curve) and semblance-based converted strain-rate (red curve) and is further illustrated in Fig. 4, where strain-rate converted accelerations, using both semblance and FK slowness, are compared with differentiated velocity for all simulated data. Comparing the residuals
between strain rate converted accelerations (panels c-d) and differentiated velocity (panel a) using semblance-derived slowness (panel e) and FK-derived slowness (panel f) demonstrates the benefit of using time-dependent slowness. The use of FK slowness produces larger residuals (panel f) than the use of semblance slowness (panel e), especially for the direct P- and S-waves (0-3 seconds). Standard deviations are thus systematically higher when using FK slowness, as seen in panel b.

To examine the performance of the semblance-based strain-rate conversion for dispersive waves (e.g., Scholte waves), this technique is applied to strain-rate records filtered at different frequency bands. For this test, noise was not added to the simulated data. An example for the same trace shown in Fig. 3 (#220) is shown in Fig. 5, where 6 different frequency bands are tested: filtered strain-rate data is shown in panels a and a comparison between converted strain-rate (red
curves) and accelerations (blue curves) is shown in panels b. Different phase velocities are measured for dispersive waves (2.5-7 seconds) at different frequency bands, while the same velocity is measured for body waves (1-2.5 seconds) (panel c). The agreement between accelerations and converted strain-rate in narrow frequency bands is only slightly better than that obtained for broadband signals, lowpass filtered at 12 Hz (top line in panels a and b), as indicated
by the standard deviations of the residuals (top right corners of panels b). The small difference between bandpass filtered (e.g., 4.8-7.6Hz in Fig. 5) and broad band (0-12Hz in Fig. 5) standard deviations of the residuals suggests that the dispersive nature of these waves has a small effect on the conversion quality. Thus, applying time-specific and frequency-specific slowness to convert broadband seismic signals is unlikely to result in a significant improvement in conversion
robustness. Also, such an approach may require intricate processing, introduce artifacts, and is highly subjective and challenging to implement. Thus, this analysis is focused on the more generic case of a broadband signal, and a bandpass limited conversion is not developed and implemented.

       The conducted analysis, and excellent agreement between simulated strain rate converted seismograms and acceleration signals, demonstrate the advantages of using the proposed slant-
stack - based approach for strain-rate to ground motion conversion. Next, the technique is implemented on earthquake recordings from three underwater DAS fibers in the Mediterranean, and the ability to determine their source parameters is demonstrated.

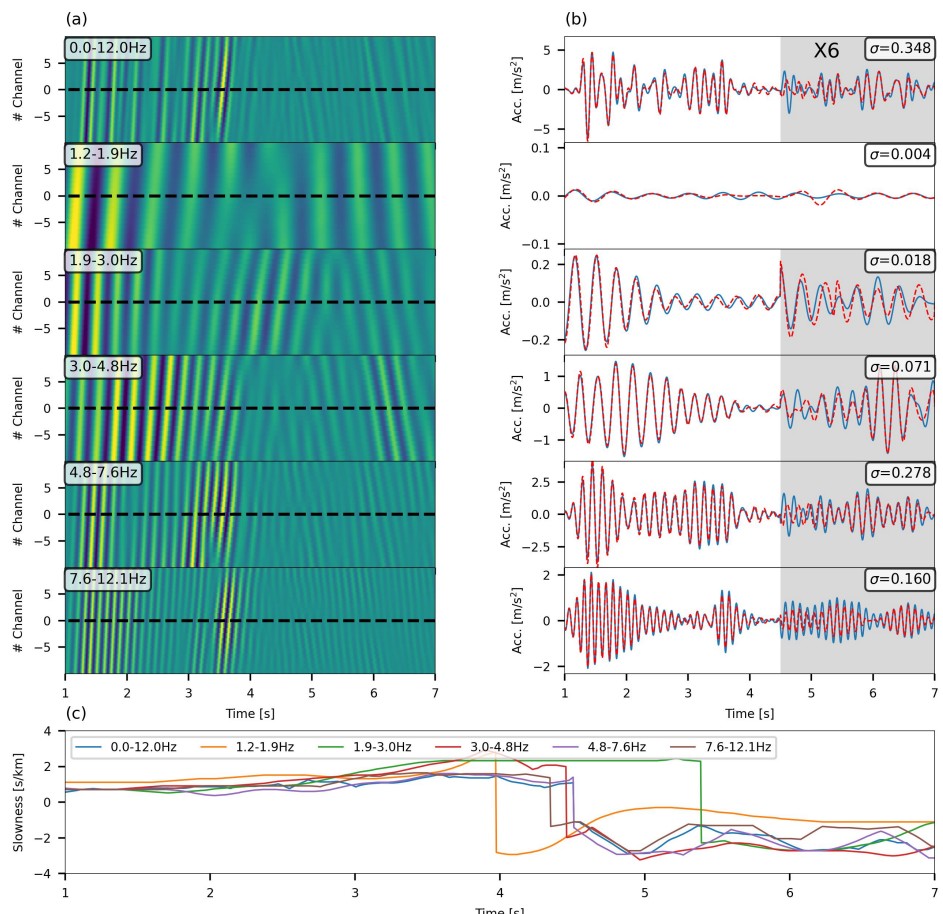

**Figure 5:** Bandpass limited conversion for channel #220 of the simulated data. Panels (a): strain-rate time series for 21 adjacent traces centered around channel #220 (dashed black line). Frequency ranges are indicated at the top left of each panel. Panels (b): accelerations, time-differentiated from simulated velocities (blue curve), compared with strain-rate converted using semblance derived slowness (red curve). Standard deviations of the residuals between converted strain-rates and time-differentiated velocities are indicated in the top right of each panel. The signals in the gray regions have been amplified by a factor of 6. Panel (c): Slowness as a function of time for different frequency bands.

## 4 Data

For a real data application of the conversion technique, 8 local earthquakes are used. These were recorded by three dark ocean-bottom telecommunication cables deployed in the

Mediterranean Sea. This dataset is identical to the one used by Lior et al. (2021) and the cables' locations, bathymetric profiles and layout are detailed there. This information, as well as earthquake data, is briefly repeated here. Data was acquired by Géoazur on two cables deployed offshore Methoni, south-west Greece, and one cable deployed offshore Toulon, South of France. In addition to DAS data, several on-land seismometers, installed near the cables, were available during the measurements. These will be later used (Sect. 6) to compare DAS and seismometer-derived source parameters. The earthquake data is detailed in Table 1, earthquakes, cables, and seismometer locations are shown in Appendix A, and magnitude and hypocentral distance distribution are shown in Fig. 6. In the latter, variations of hypocentral distance correspond to the different analyzed fiber segments. The earthquake DAS records used in this study are dominated by scattered Scholte-waves, with velocities between 100 and 400 m s$^{-1}$ (Lior et al., 2021).

The two cables deployed offshore Greece are used for the HCMR (Hellenic Centre for Marine Research) and NESTOR (Neutrino Extended Submarine Telescope with Oceanographic Research) projects. The acquisitions were conducted on 18-19 and 19-25 April 2019 on the HCMR an NESTOR cables, respectively. A Febus A1 DAS interrogator was used, measuring strain-rate signals. Data was sampled at 6 and 5 ms for HCMR and NESTOR, respectively, and gauge length and spatial sampling were both set to 19.2 m for both cables. These records amount to 688 and 1365 equally spaced channels for the 13.2 and 26.2 km long HCMR and NESTOR cables, respectively. In addition, 2 seismometers installed near the on-land end of the fibers were available during part of these campaigns, METN and METS.

The cable deployed offshore Toulon is used for the MEUST-NUMerEnv project

**Table 1:** Earthquakes used in this study.

| Cable name | Origin time (UTC) | Magnitude (local) | Location (latitude, longitude, depth[km]) | catalog |
|---|---|---|---|---|
| NESTOR | 22/04/2019 19:26:06 | 3.3 | 37.4185, 20.6897, 11.0 | Athens University |
| | 23/04/2019 17:29:40 | 3.6 | 37.7753, 20.7658, 7.0 | Athens University |
| | 21/04/2019 22:11:47 | 2.0 | 36.8335, 22.0382, 2.0 | Athens University |
| | 23/04/2019 19:25:51 | 2.6 | 37.2528, 21.4593, 9.0 | Athens University |
| HCMR | 18/04/2019 21:44:42 | 3.7 | 37.57, 20.66, 8.0 | EMSC |
| | 19/04/2019 03:30:19 | 2.6 | 37.1523, 20.6662, 1.0 | Athens University |
| MEUST | 19/07/2019 21:16:57 | 2.6 | 44.374, 6.913, 2.6 | Géoazur |
| | 21/07/2019 23:01:58 | 2.4 | 42.516, 5.143, 2.0 | Géoazur |

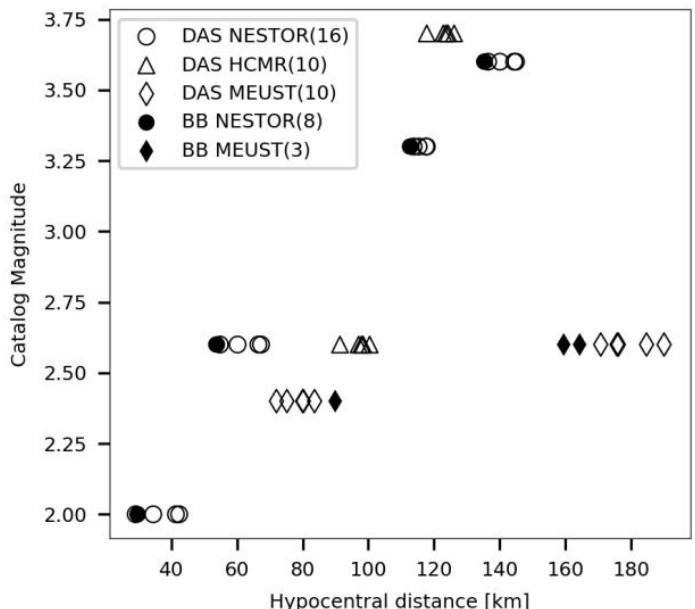

**Figure 6:** Earthquake catalog magnitude ($M_L$) as a function of hypocentral distances for the earthquakes in this study (Table 1). Hypocentral distances are measured to specific cable segments or broadband (BB) sensors. Solid and empty symbols correspond to seismometer and DAS data, respectively. Data for NESTOR, HCMR and MEUST are indicated by circles, triangles and diamonds, respectively, and the number of data points is indicated in parentheses in the legend.

(Mediterranean Eurocentre for Underwater Sciences and Technologies - Neutrino Mer Environnement) (Lamare, 2016) and was previously harnessed for DAS measurements by Sladen et al. (2019). The acquisition was conducted on 11-31 July 2019 using a chirped-pulse hDAS interrogator developed by Aragon Photonics, measuring strain signals (Pastor-Graells et al., 2016; Fernandez-Ruiz et al., 2019; Williams et al., 2019). Data was sampled at 10 and 2 ms for the first and last 10 days of the campaign, respectively, and gauge length and spatial sampling were both set to 10 m. These records amount to 4480 equally spaced channels for the 44.8 km long cable. In addition, 2 seismometers were installed near the on-land end of the fiber, POSAN and POSAS.

## 5 Application to DAS recorded earthquakes

To demonstrate the performance of the proposed conversion approach, DAS strain (rate) earthquake signals are converted to ground velocities (accelerations). For each of the three cables,

short fiber segments that exhibit coherent and continuous waveform recordings are chosen. Each section contains 29 traces, that are filtered between 1 and 5 Hz. For different applications, a different filter may be used, as demonstrated in the next section. Filtered signals are converted to ground velocities or accelerations using fiber segments of ~380 m (L=10 for NESTOR and HCMR and L=19 for MEUST). Compared with simulated data, longer segments are used in order to resolve faster seismic phases and longer wavelengths. For the FK transform, 2 s data intervals were used. Applying the conversion to DAS recorded earthquakes highlights body-wave arrivals, since these fast waves exhibit higher converted amplitudes compared with later arriving scattered waves and presignal ambient noise. Figure 7 shows an example of strain conversion to ground velocities for an $M_L$2.6 earthquake recorded by the MEUST cable at a hypocentral distance of 185 km, and Fig. 8 shows an example of strain-rate conversion to ground accelerations for an $M_L$3.6 earthquake recorded by the NESTOR cable at a hypocentral distance of 140 km. In Fig. 7, mostly direct S-waves are shown, exhibiting unilateral wave propagation (panels a-c). The apparent velocity of the direct S-waves (1.2-2.2 seconds) is determined to be 2 and 1 km s$^{-1}$ using semblance and FK-derived slowness, respectively, while later arriving waves travel at ~400 and ~370 m s$^{-1}$ using semblance- and FK-derived slowness, respectively (panel c). Thus, as observed for the simulated data (Sect. 3), direct waves (panels b and d) exhibit higher converted velocity amplitudes compared to later phases. This is visualized by comparing the color-codes in panels (a) and (b). Figure 8 shows both P- and S-waves, as well as presignal noise. In this example, scattered waves, and thus bilateral wave propagation, dominate the measurements (Lior et al., 2021), and several slowness sign flips are evident. The apparent velocity of first arriving S-waves (24-25 seconds) is 1.3 km s$^{-1}$ and 800 m s$^{-1}$ using semblance- and FK-derived slowness, respectively, while the average apparent velocity is 750 and 530 m s$^{-1}$, using semblance- and FK-derived slowness, respectively, resulting in higher converted acceleration amplitudes for the direct waves. Ocean-bottom presignal noise is dominated by instrument related effects (e.g., Lior et al., 2021; Costa et al., 2019) and ambient noise. These signals are characterized by low apparent velocities, which results in low acceleration amplitudes, and facilitates easy identification of the initial P-waves, subject to SNR conditions.

Next, the ability to invert for the source parameters, i.e., seismic moment, corner frequency and stress drop, is examined by converting strain (rate) records for predefined P- and S-wave intervals and fitting their spectra with an earthquake source model.

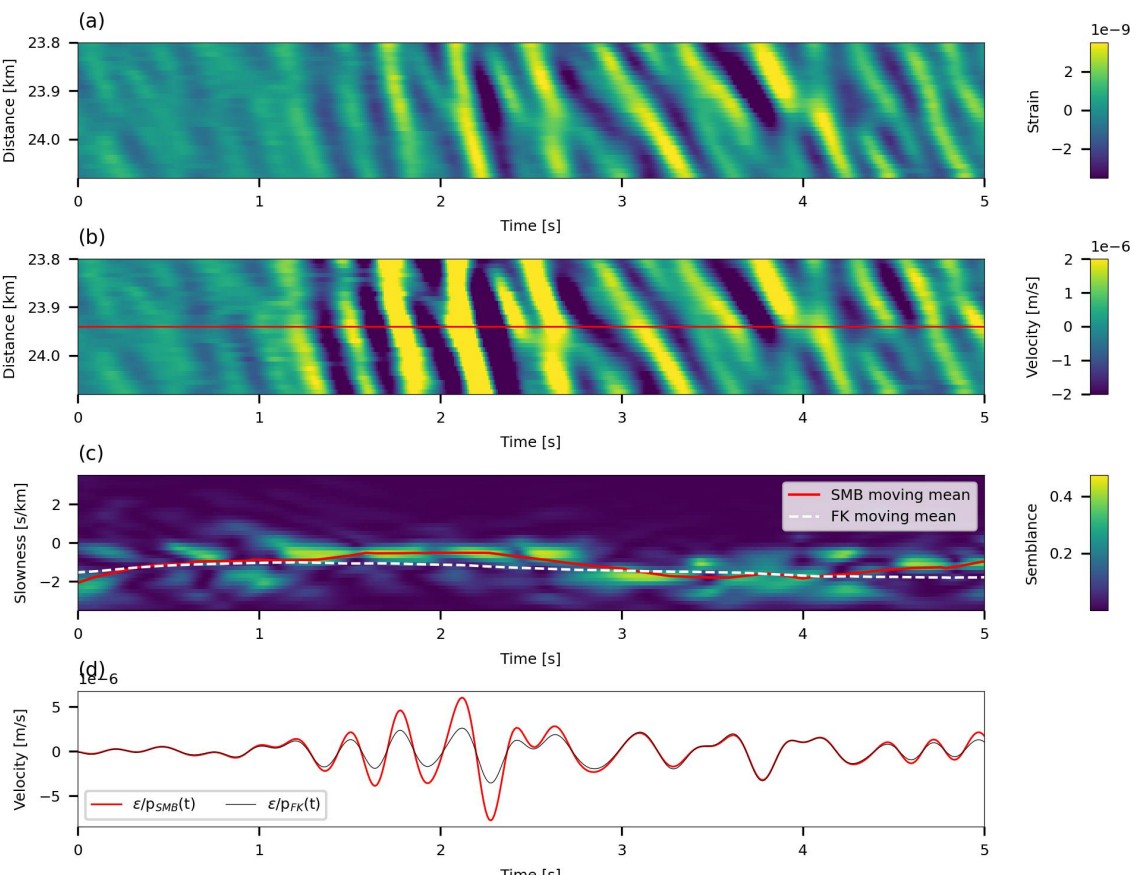

**Figure 7:** Slant-stack conversion for traces between 23.8 and 24.1 km along the MEUST cable of an $M_L 2.6$ recorded at a hypocentral distance of 185 km. 5 seconds around the direct S-wave arrival are shown. Panel (a): strain time series for 29 adjacent traces. Panel (b): Strain converted to velocities for all 29 traces. Panel (c): Slowness as a function of time color coded by semblance values for the middle channel, indicated by the red line in panel (b). Red and dashed white curves correspond to semblance and FK derived smoothed slowness, respectively. Panel (d): Velocity converted using semblance and FK derived slowness are indicated by red and black curves, respectively.

## 6 Implications for source parameter inversion

Seismic moment, source corner frequency, and stress drop are determined by fitting converted earthquake DAS signals with an earthquake source model. The source model chosen is

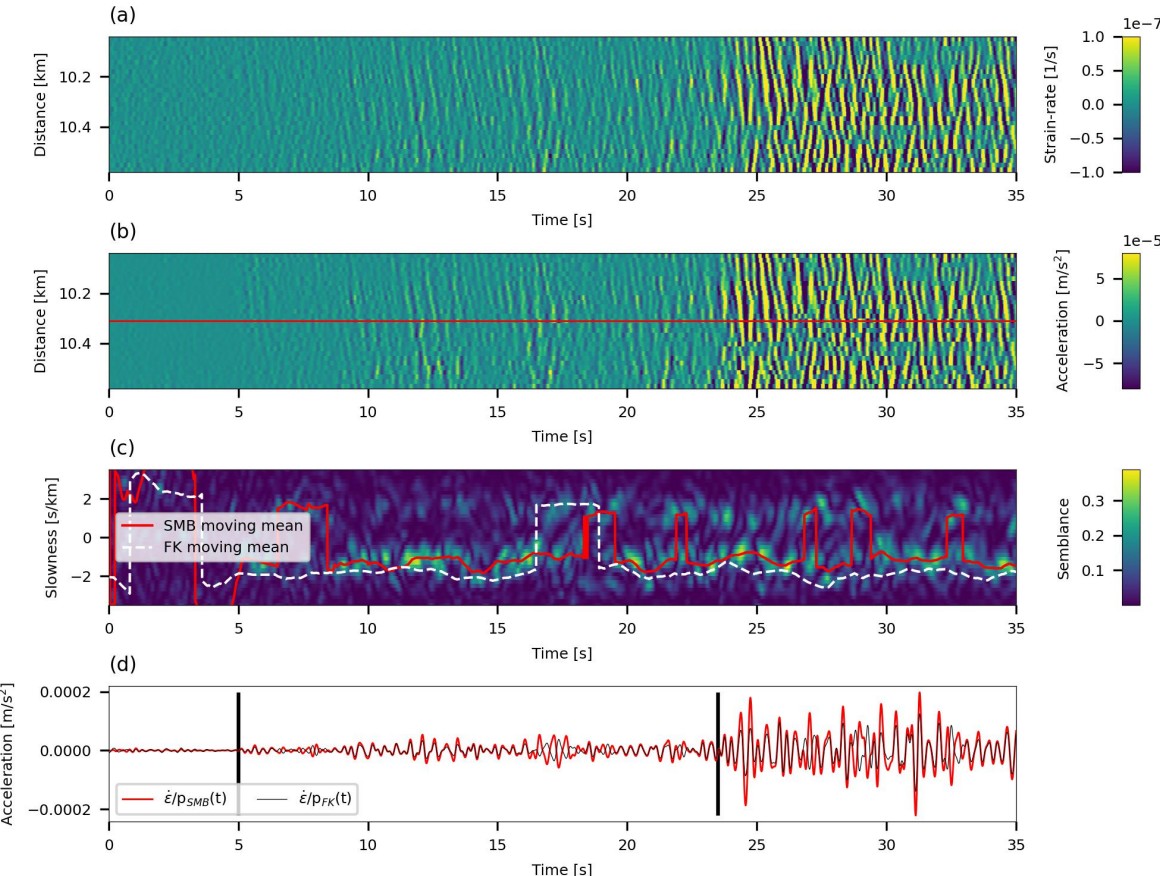

**Figure 8:** Slant-stack conversion for traces between 10.05 and 10.6 km along the NESTOR cable of an $M_L3.6$ recorded at a hypocentral distance of 140km. Both P- (~5 seconds) and S-wave (~23.5 seconds) arrival are shown. Panel (a): strain-rate time series for 29 adjacent traces. Panel (b): Strain-rate converted to accelerations for all 29 traces. Panel (c): Slowness as a function of time color coded by semblance values for the middle channel, indicated by the red line in panel (b). Red and dashed white curves correspond to semblance and FK derived smoothed slowness. Panel (d): Acceleration converted using semblance and FK derived slowness are indicated by red and black curves, respectively.

the commonly used omega squared model (e.g., Brune, 1970, Madariaga 1976; Sato and Hirasawa, 1973) which describes the far-field body wave radiation for ground displacements, velocities and accelerations. The model is fit to the data via the single-step inversion of Lior and Ziv (2018). This

approach is advantageous as model fitting is done in the time-domain, circumventing the time- to frequency-domain transformation and avoiding many spectral model fitting intricacies.

## 6.1 Source model

For ground displacements, velocities and accelerations, the omega-squared model subject to high frequency attenuation (Anderson and Hough, 1984) is given by:

$$\Omega(f) = \frac{\Omega_0}{1+(f/f_0)^2} exp(-\pi \kappa f), \tag{3a}$$

$$\dot{\Omega}(f) = 2\pi f \frac{\Omega_0}{1+(f/f_0)^2} exp(-\pi \kappa f), \tag{3b}$$

and

$$\ddot{\Omega}(f) = (2\pi f)^2 \frac{\Omega_0}{1+(f/f_0)^2} exp(-\pi \kappa f), \tag{3c}$$

respectively, where $\Omega_0$ is the low-frequency displacement spectrum plateau, $f$ is frequency, $f_0$ is the source corner frequency and $\kappa$ is the attenuation parameter. The latter can be expressed as an attenuation corner frequency as $f_\kappa = 1/(\pi \kappa)$ (Eq. 4 of Lior and Ziv, 2018). The spectral parameters

$\Omega_0$ and $f_0$ correspond to the seismic moment, $M_0$, and stress drop, $\Delta \tau$, as:

$$M_0 = \Omega_0 \frac{4\pi \rho C^3 R}{U_{\varphi \theta} F_s}, \tag{4a}$$

$$\Delta \tau = \frac{7}{16} M_0 \left(\frac{f_0}{kC_S}\right)^3, \tag{4b}$$

where $\rho$ is the density at the source, $C$ is the wave velocity at the source ($C_P$ and $C_S$ for P- and S-waves, respectively), $R$ is the hypocentral distance, $U_{\varphi \theta}$ is the radiation pattern, $F_s$ is the free-surface effect, and $k$ is a constant which depends on the wave type and rupture speed (Madariaga, 1976). Equation (4b) applies to a circular crack (Eshelby, 1957) expanding isotropically at constant

rupture speed. Parameter tuning is set as follows: $\rho$=2600 kg m$^{-3}$, $C_P$=5333 m s$^{-1}$, $C_S$=3200 m s$^{-1}$ (e.g., Lior and Ziv, 2020), $U_{\varphi \theta}$ equals 0.52 and 0.63 for P- and S-waves, respectively (Aki and Richards, 1980), $F_s$=2, and $k$ equals 0.32 and 0.21 for P- and S-waves, respectively, corresponding to a rupture speed of $0.9C_S$ (Madariaga, 1976). Using $F_s = 1.7$ instead of $F_s = 2$, as is sometimes used for ocean-bottom applications (e.g., Webb, 1998), will reduce magnitudes estimates by

~0.047, a minute difference compared to magnitude uncertainties, as shown in the next subsection.

## 6.2 From strain (rate) to source parameters

The spectral parameters are determined via the single-step inversion of Lior and Ziv (2018), which resolves $\Omega_0$, $f_0$ and $\kappa$ using the time-domain signals, circumventing the time- to frequency-domain transformation, required by most source parameter inversion methods. The approach is fully detailed in Lior and Ziv (2018) and briefly summarized in Appendix B.

DAS strain (rate) data is converted to ground velocity (acceleration) for manually chosen P- and S-wave windows and source parameters are resolved. The procedure of determining the frequency band of interest, strain (rate) to ground motions conversion, and model fitting is demonstrated in Fig. 9 for the S-waves of an $M_L 3.6$ earthquake, recorded between 20.95 and 21.5 km along the NESTOR cable, at a hypocentral distance of 145 km. First, for the 29 traces composing each cable segment, strain (rate) signal and noise amplitude spectra (AS) are calculated, resampled (following the procedure described in McNamara and Buland, 2004) and stacked for signal and presignal time-windows of equal length (dashed black and solid gray curves in panel a, respectively). The analyzed presignal noise is that recorded 2 minutes before the signals. The bandwidth for which frequency-specific SNR (signal AS(f) / noise AS(f)) is larger than 2 is used for subsequent analysis (solid black curve in panel a). If less than 3 discrete frequencies have high SNR, the recording is disregarded. To fully preserve the frequency-band of interest, the filter's lower and upper corner frequencies are slightly decreased and increased by factors of $10^{-0.2}$ and $10^{0.2}$, respectively. Strain (rate) signals are then filtered (panel b), converted to ground velocities (accelerations) (panel c), and differentiated and/or integrated to obtain ground displacements, velocities and accelerations. Following each differentiation/integration, the forementioned filter is applied. The signals' RMS are calculated in the time-domain and source parameters are determined as detailed in Appendix B. An example of the single-step inversion's results is shown in Fig. 9d where the best fitting model is plotted using Eq. (3c) and compared with observed stacked acceleration spectra, and Fig. 9e shows the best fitting parameter combination ($\Omega_0$, $f_0$ and $\kappa$) indicated by a red star, in $log(f_0) - log(f_\kappa)$ space. Color code in panel e corresponds to the best fitting $\Omega_0$ (for each $log(f_0) - log(f_\kappa)$ combination) and the contours correspond to the objective function's value. The solution exhibits a high degree of trade-off between the values of $f_0$ and $\kappa$ (panel e), yet good agreement is found between observed (black curve) and modeled (blue curve) spectra (panel d).

To further compare the performance of FK- and semblance-based slowness, magnitude estimates are compared. Figure 10 plots the difference between magnitudes estimated following the FK- and semblance-based conversion schemes as a function of catalog magnitude. Magnitudes

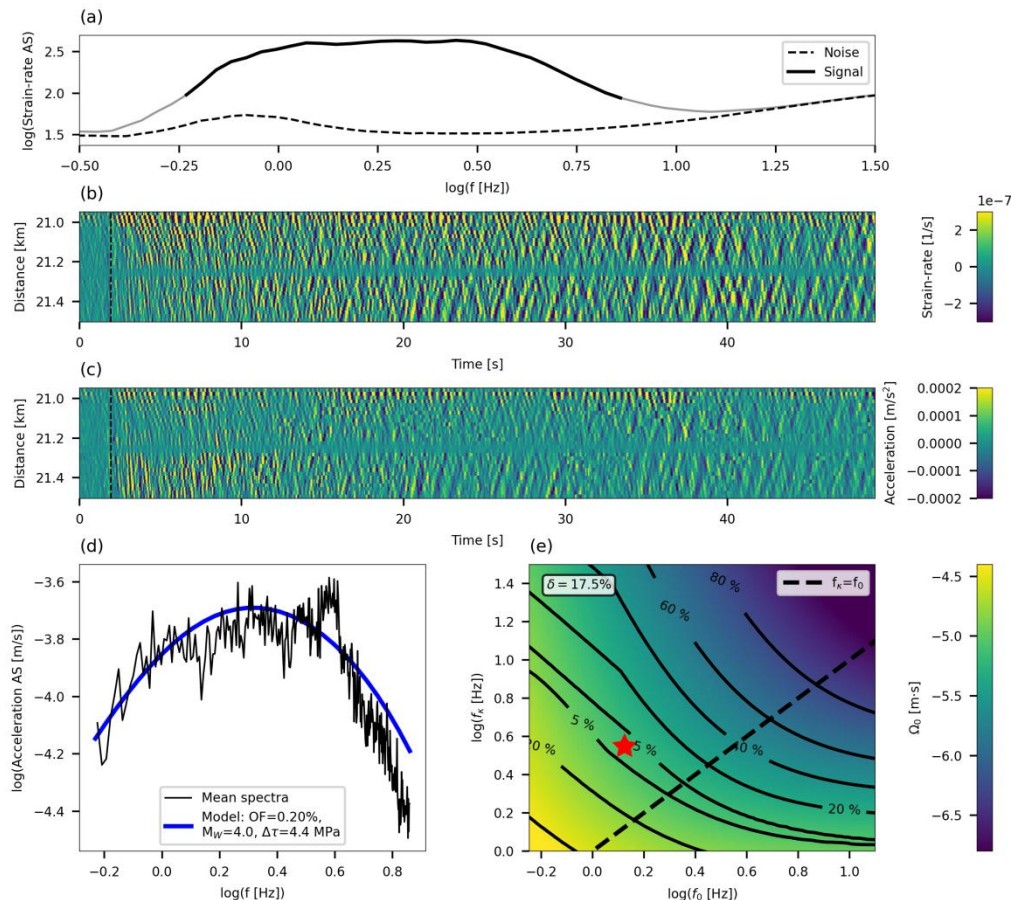

**Figure 9:** Source parameter inversion procedure for an $M_L 3.6$ recorded at a hypocentral distance of 145km between 20.95 and 21.5 km along the NESTOR cable. Panel (a): stacked resampled signal and noise amplitude spectra (AS) are indicated by solid gray and dashed black curves, respectively. High-SNR signal (frequency specific SNR>2) is indicated by a solid black curve. Panel (b): strain-rate time series for 29 adjacent traces. Panel (c): Strain-rate converted to accelerations using the semblance approach for all 29 traces. Panel (d): Stacked acceleration AS and best fitting earthquake model are plotted in black and blue curves, respectively. Panel (e): Contour diagram of the inversion's objective function in $log(f_0) - log(f_\kappa)$ space with color code corresponding to the best fitting $\Omega_0$. The uncertainty parameter, $\delta$, is indicated in the top-left corner, and the dashed black line indicates $f_0 = f_\kappa$.

estimated for data converted using FK-derived slowness are generally lower than those resolved using semblance-derived slowness. This trend is expected given the lower apparent velocities

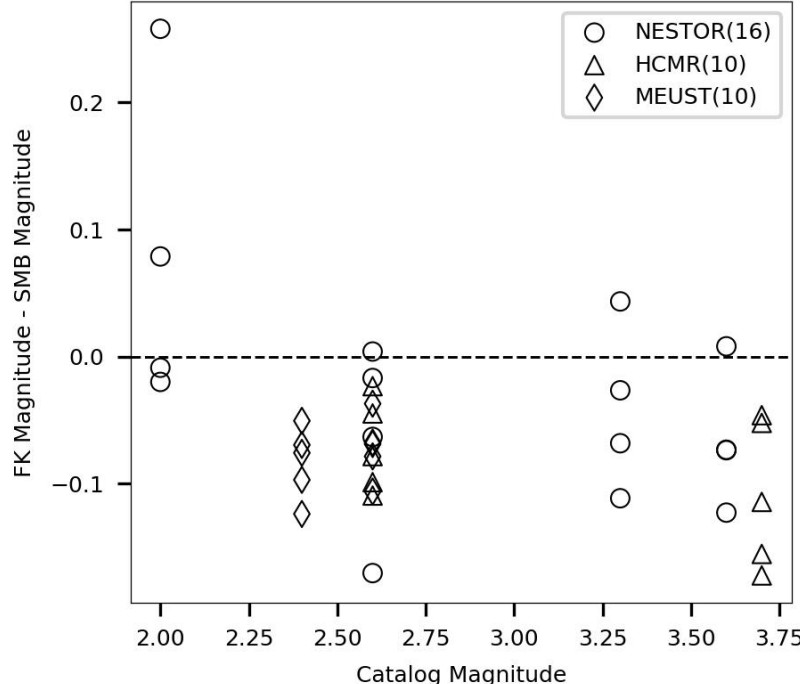

**Figure 10:** Magnitude discrepancies: Magnitudes estimated following the FK based slowness conversion minus magnitudes estimated following the semblance based conversion, plotted as a function of catalog magnitude. Data for NESTOR, HCMR and MEUST are indicated by circles, triangles and diamonds, respectively, and the number of data points is indicated in parentheses in the legend.

determined via the FK-based approach, as previously shown for both simulated (Fig. 3) and observed (Fig. 7 and 8) earthquakes.

Moment magnitudes, stress drops and corner frequencies resolved using DAS semblance-
based converted data are found to be in good agreement with those estimated on adjacent on-land broadband seismometers. Since the DAS fiber and the seismometers are not colocated, their estimated source parameters are not expected to be in perfect agreement. DAS P- and S-wave magnitude estimates are plotted as a function of average S-wave seismometer magnitudes in Fig. 11, and DAS S-wave $f_0$ and $\Delta\tau$ estimates are plotted as a function of average S-wave seismometer
obtained parameters in Fig. 12. The low SNR conditions observed for P-waves, i.e. the narrow available frequency-band, did not allow for robust estimates of $f_0$ and $\Delta\tau$, which are thus not shown. A similar comparison with catalog magnitude is shown in Fig. 13, noting that local and

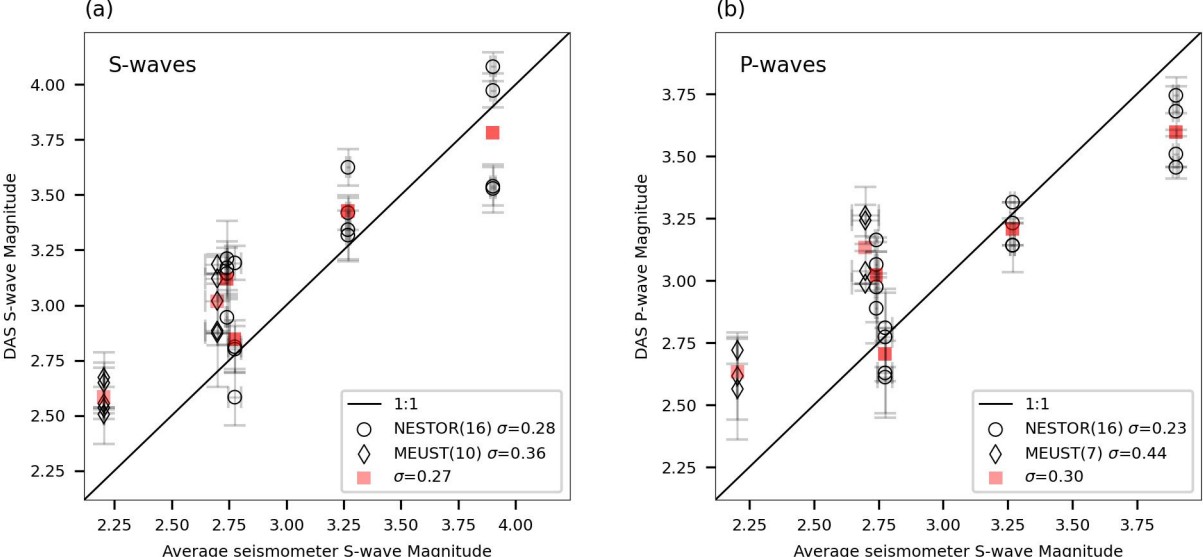

**Figure 11:** Comparison between DAS and seismometer magnitude estimates. Moment magnitudes estimated using DAS recorded S-waves (panel a) and P-waves (panel b) are plotted as a function of average magnitudes obtained using seismometer records. DAS event averaged magnitudes are plotted in red squares. The black curve is a 1:1 line. The number of data points and standard deviations to magnitude residuals are indicated in the legends. DAS magnitude errors are the standard deviations for magnitudes determined for each individual seismogram in the analyzed fiber segment, while seismometer errors are the standard deviations for single seismometer estimates.

moment magnitudes may differ for small earthquakes (Deichmann, 2006). DAS parameter errors were calculated as the standard deviations of parameters determined for each individual seismogram in the analyzed fiber segment, while seismometer errors are the standard deviations of single seismometer estimates, when available. Data for the HCMR cable is not shown since seismometer gain was unavailable.

## 7 Discussion

The comparison between semblance-derived slowness and FK-derived slowness for strain (rate) to ground motions conversion reveals several advantages favoring the semblance-based approach. Unlike semblance analysis, whose implementation and interpretation is simple and objective, FK analysis introduces considerable subjectivity into the slowness determination

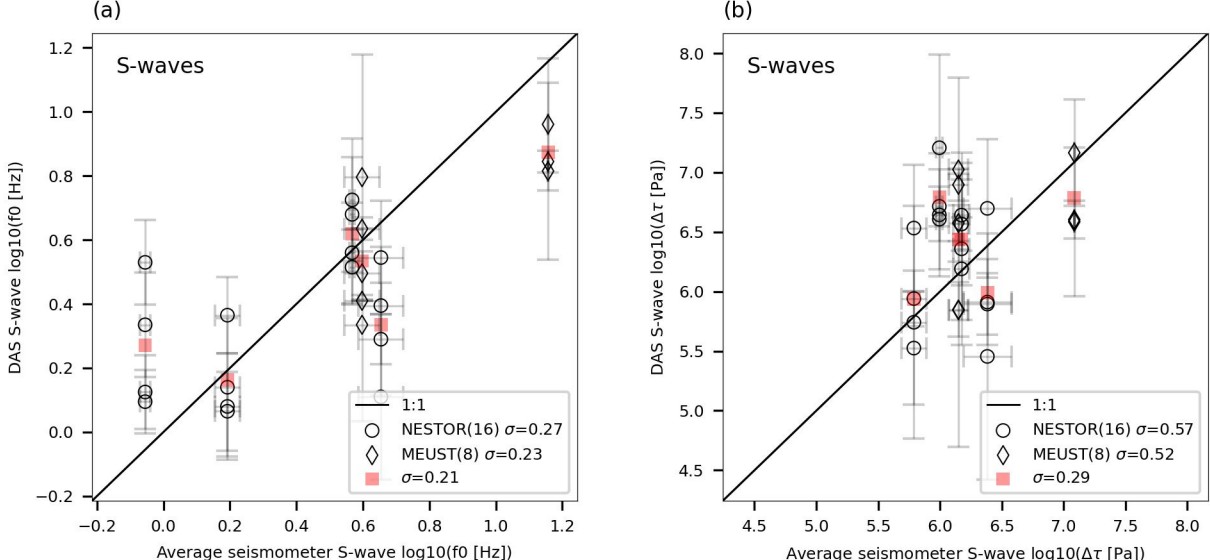

**Figure 12:** Comparison between DAS and seismometer source corner frequencies and stress drops. $log(f_0)$ (panel a) and $log(\Delta\tau)$ (panel b) estimated using DAS recorded S-waves are plotted as functions of average parameters obtained using seismometer records. DAS event averaged parameters are plotted in red squares. The black curve is a 1:1 line. The number of data points and standard deviations to parameter residuals are indicated in the legends. DAS parameter errors are the standard deviations for parameters determined for each individual seismogram in the analyzed fiber segment, while seismometer errors are the standard deviations for single seismometer estimates.

procedure since interpreting an FK image may be done in various ways. The FK analysis generally yielded lower velocities owing to its poor slowness resolution, as observed for both simulated and
recorded earthquakes. For these reasons, when comparing simulated accelerations and converted strain-rates, a significantly better agreement was obtained using the semblance-based approach. The ground motion conversion difference between FK- and semblance-based approaches is most pronounced for direct arrivals, i.e. P- and S-waves. Thus, the semblance-based conversion is particularly advantageous for EEW, when short duration, relatively fast propagating waves, are
required for speedy and reliable source parameter estimation.

A significant hinder when using DAS is the mechanical coupling between the fiber and the solid Earth. This issue is particularly troublesome when standard telecommunication fibers are used, specifically those deployed underwater, as their coupling quality is often unknown and may prevent reliable seismic monitoring (e.g., Sladen et al., 2019; Lior et al., 2021). Here, specific

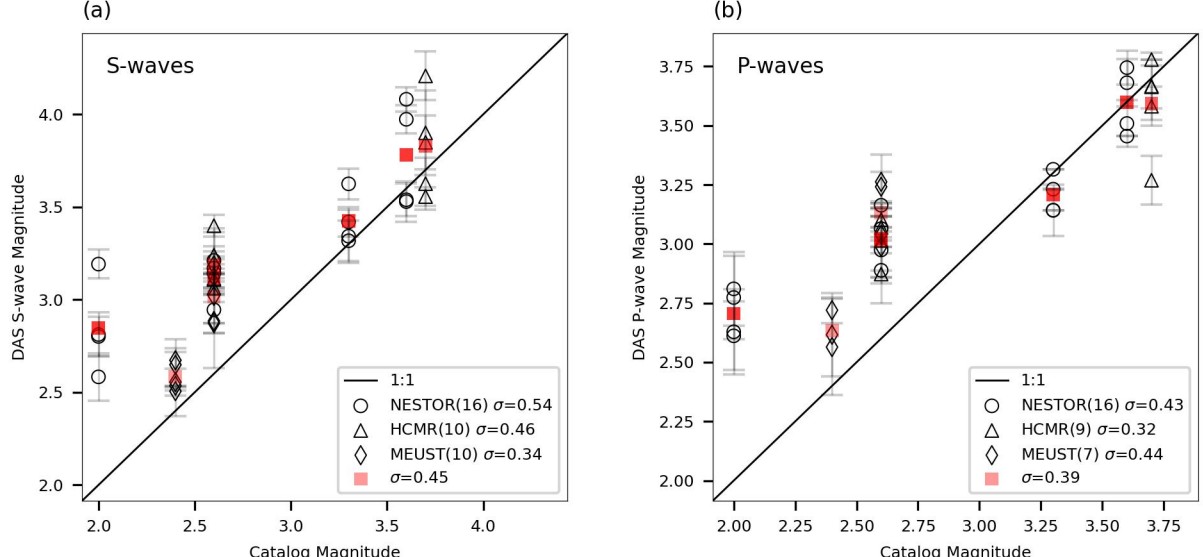

**Figure 13:** Comparison between DAS and catalog magnitude estimates. Moment magnitudes estimated using DAS recorded S-waves (panel a) and P-waves (panel b) are plotted as a function of catalog magnitudes. DAS event averaged magnitudes are plotted in red squares. The black curve is a 1:1 line. The number of data points and standard deviations to magnitude residuals are indicated in the legends. DAS magnitude errors are the standard deviations for magnitudes determined for each individual seismogram in the analyzed fiber segment.

425 segments for which recording quality is sufficiently uniform were manually identified. The effect of coupling along these limited segments is quantified by considering the signals' average absolute strain-rate amplitudes in decibels (dB), plotted in Fig. 14 for all traces, segments and cables. In Fig. 14, only earthquakes recorded at hypocentral distances longer than 80 km are plotted, to ensure slow propagation-related amplitude changes along the fiber. The variabilities are generally small,

430 limited to 2-3 dB with several exceptions of ~8 dB. These mostly minor deviations, along with the small DAS magnitude uncertainties (vertical errorbars in Fig. 11 indicate standard deviations of magnitudes resolved independently for each DAS channel) indicate that these segments display sufficiently uniform coupling for ground motion conversion and source parameter estimation. Moreover, the fact that even for non-uniform coupled cables as those used here, sufficiently

435 uniform coupling is observed, even if in limited segments, demonstrates the potential of underwater fibers for reliable source parameter estimation.

   The proposed slant stack conversion approach relies on the ability to resolve the phase velocity of a single plane wave at every time instance. However, seismic records are often

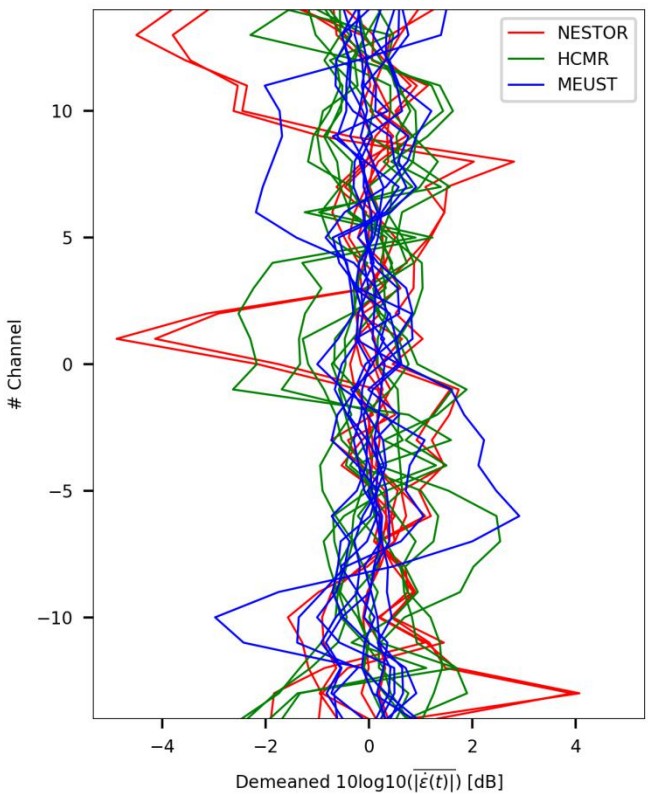

**Figure 14:** Signal amplitude variability for all analyzed fiber segments for the 3 cables. Number of channel along the limited segments (29 traces) as a function of demeaned $10log_{10}(X)$ with $X$ being the average absolute strain-rate amplitudes. Data for NESTOR, HCMR and MEUST are plotted in red, green and blue, respectively.

dominated by several waves, which may be dispersive (e.g. surface waves), characterized by
different velocities, incidence angles, and exhibit complex propagation, scattering and interference patterns. The analysis on simulated data in Sect. 3 demonstrates that when a single plane wave is considered, converted strain-rates are in excellent agreement with acceleration waveforms, while when two opposing plane waves interfere, the conversion's robustness is decreased (e.g., Fig. 3). Careful filtering of DAS signals needs to be applied to isolate specific plane waves from DAS
earthquake records, as demonstrated in part in Fig. 5, yet conversion errors may still result from inadequate slowness resolution, incoherent plane waves, and noise. The effect of noise is seen in Sect. 3 where synthetic P-waves suffer from low SNR, which results in lower than expected apparent velocities, and a slightly reduced conversion quality. The amplitudes of DAS P-waves,

which arrive at near-vertical incidence angles with respect to horizontal fibers, are especially low
since they are reduced by a factor of squared cosine of incidence angle (e.g. Mateeva et al., 2014).
Further consideration of these issues is beyond the scope of this manuscript. In-spite of these
complexities, converted DAS signals allow for reliable magnitude estimation, demonstrating the
robustness of the conversion procedure.

Resolving source corner frequencies, and thus stress drops, for small and/or distant
earthquakes in unfavorable SNR conditions, is a challenging task since high frequency source
effects, i.e., $f_0$, are masked by high frequency attenuation, i.e., $f_\kappa$ (e.g., Lior and Ziv, 2018). To
address this issue, Lior and Ziv (2018) introduced an uncertainty parameter, $\delta$, a quality control
measure for the ambiguity between best fitting $f_0$ and $\kappa$. The value of $\delta$ is proportional to the area
enclosed within the 5% contour in $log(f_0) - log(f_\kappa)$ space (e.g., Fig. 9e). Lior and Ziv (2018) found
that solutions with high $\delta$ values (typically > 6%) usually exhibit a high degree of ambiguity
between $f_0$ and $\kappa$. For such cases, they implemented a two-step inversion approach, which
consists of determining a station, or in this case fiber segment, specific $\kappa$ using low $\delta$ solutions and
repeating the inversion for $\Omega_0$ and $f_0$. However, since for the data used in this study only few low $\delta$
solutions were obtained, this technique is not implemented here. Thus, corner frequencies and
stress drops are not well constrained. The standard deviations to parameter residuals (Fig. 12) are
within-event variabilities between the estimates of specific DAS segments and seismometers. That
these values are only slightly higher than within-event variabilities reported by Lior and Ziv (2018)
suggests that in-spite of the inability to reliably determine these parameters, DAS and
seismometer-derived $f_0$ and $\Delta\tau$ are found to be in good agreement (Fig. 12).

Even in cases where $f_0$ and $\kappa$ (and thus $\Delta\tau$) may not be well constrained, $\Omega_0$, and thus
moment magnitudes, are reliably determined (Lior and Ziv, 2018). This is visualized in Fig. 9e,
where $\Omega_0$ values (color code) are generally sub-parallel to the OF contours.

Magnitudes are reliably determined for both P- and S-waves (Fig. 11) in spite of the reduced
sensitivity of horizontal fibers to transverse deformations, as expected for P-waves. Recorded
deformation amplitudes are modulated by $cos^2\theta$, where $\theta$ is the wave's incidence angle with
respect to the fiber's axis (e.g., Ajo-Franklin et al., 2019; Kuvshinov, 2016; Mateeva et al., 2014;
Papp et al., 2017; Yu et al., 2018), thus, DAS measurements are mostly sensitive to deformations
along the fibers' axis, i.e. elongation and compression. Since direct P-waves are expected to arrive
at near vertical incidence angles, they would not induce significant deformations: while direct S-
wave arrivals are clearly identified for several fiber segments (e.g., Fig. 7), direct P-waves are not.
However, analyzed DAS records are dominated by low velocity waves, following both direct P- and

S-wave arrivals (e.g., Fig. 8). These are scattered P- and S-waves, propagating in a variety of horizontal directions (Lior et al., 2021), which are easily measured on horizontal fibers, and used here to infer source parameters. That earthquake magnitudes, determined by DAS measurements of scattered waves, are in close agreement with both catalog magnitudes (Fig. 13) and seismometer-derived magnitudes (Fig. 11) indicates that these waves reliably represent the source characteristics and may be used for source parameter inversion.

The ability to infer source parameters using P-waves recorded on horizontal fibers is key for harnessing DAS, specifically using underwater fibers, for EEW. To this end, the proposed algorithm will need to be adapted for real-time performance. The goal of EEW systems is to robustly and rapidly predict ground shaking intensities, an objective that is typically achieved by estimating earthquake source properties in real-time (e.g., Allen and Melger, 2019; Lior and Ziv, 2020). To this end, and in order to issue ground shaking alerts as early as possible, seismic observations should be obtained at close proximity to earthquake epicenters, and source parameters should be estimated using both P- and S-waves. Since many of the most hazardous earthquakes on Earth occur at subduction zones, and therefore underwater, the ability to determine source parameters using both P- and S-wave recorded by ocean-bottom DAS, will significantly improve the performance of EEW system for underwater earthquakes and enhance hazard mitigation capabilities.

## 8 Conclusions

In this study, the ability to convert DAS strain (rate) signals to ground motion records and resolve earthquake source parameters is demonstrated. An algorithm for DAS data to ground motion conversion is presented: apparent phase slowness is determined at every time instance using semblance-based local slant-stack transform, and used to convert strain (rate) to ground velocities (accelerations). The algorithm is successful at resolving the apparent velocities of different seismic phases. Validation using simulated waveforms reveals excellent agreement between simulated accelerations and converted strain-rate signals even in the presence of correlated noise and propagation direction variations. Application of the algorithm to 8 earthquakes recorded by ocean bottom DAS fibers in the Mediterranean Sea highlights fast waves (body-waves) since they exhibit high converted ground motion amplitudes compared with low-velocity scattered waves and presignal ambient noise. Earthquake magnitudes and stress drops were determined for P- and S-waves using the single-step approach of Lior and Ziv (2018), circumventing the time- to frequency-domain transformation typically required for moment and corner frequency estimation.

Close agreement is observed between source parameters determined using on-land broadband
seismometers and ocean-bottom DAS, even when source corner frequencies and stress drops are
not well constrained due to significant high frequency attenuation. This ability to resolve
earthquake magnitudes using P-waves recorded by horizontal ocean-bottom fibers is key for
implementing DAS for EEW. The algorithm for strain (rate) conversion may be adapted for real-time
applications and used in conjunction with real-time source parameter determination schemes (e.g.
Lior and Ziv, 2020) for a DAS-based EEW system. Harnessing DAS for EEW, specifically using
ocean-bottom fibers, will significantly improve hazard mitigation capabilities for underwater
earthquakes and tsunami earthquakes.

**Appendix A**

The cables, earthquakes and seismometer locations are shown in the maps in this section.

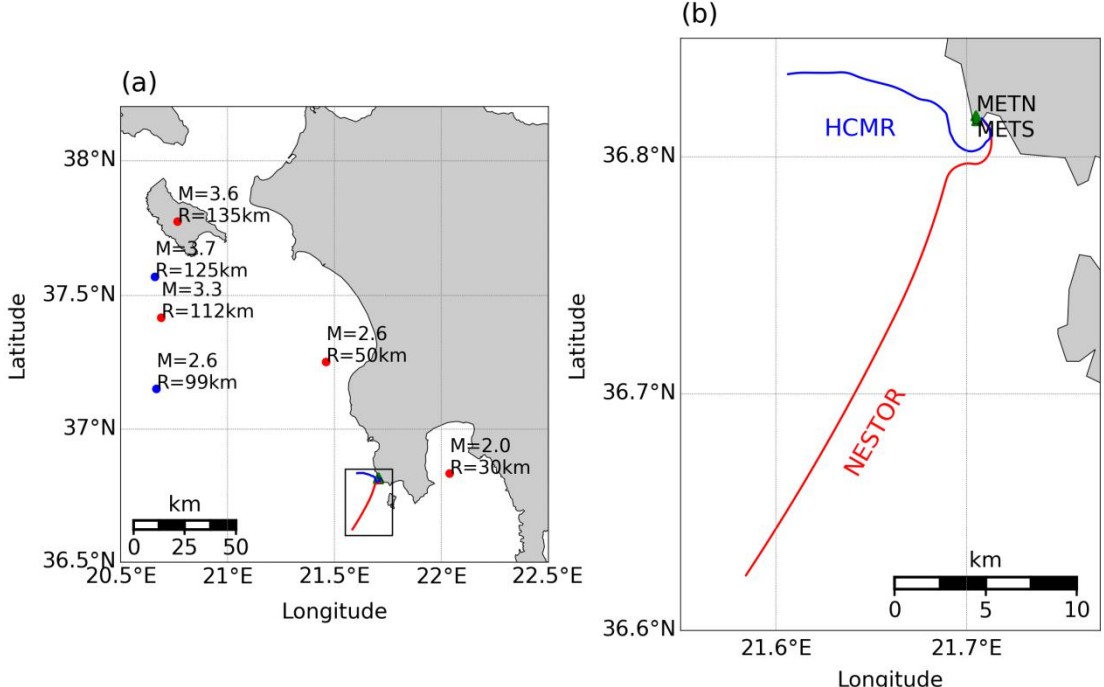

**Figure A1:** Map of earthquake, seismometer and cable locations in Greece. The NESTOR and
HCMR cables (lines) and their recorded earthquakes (circles) are indicated in red and blue,
respectively. Panel (a): catalog magnitudes and hypocentral distances are indicted near
earthquake locations. Panel (b) shows the region marked by a black rectangle in panel (a), with
cable layout and broadband seismometers (green triangles).

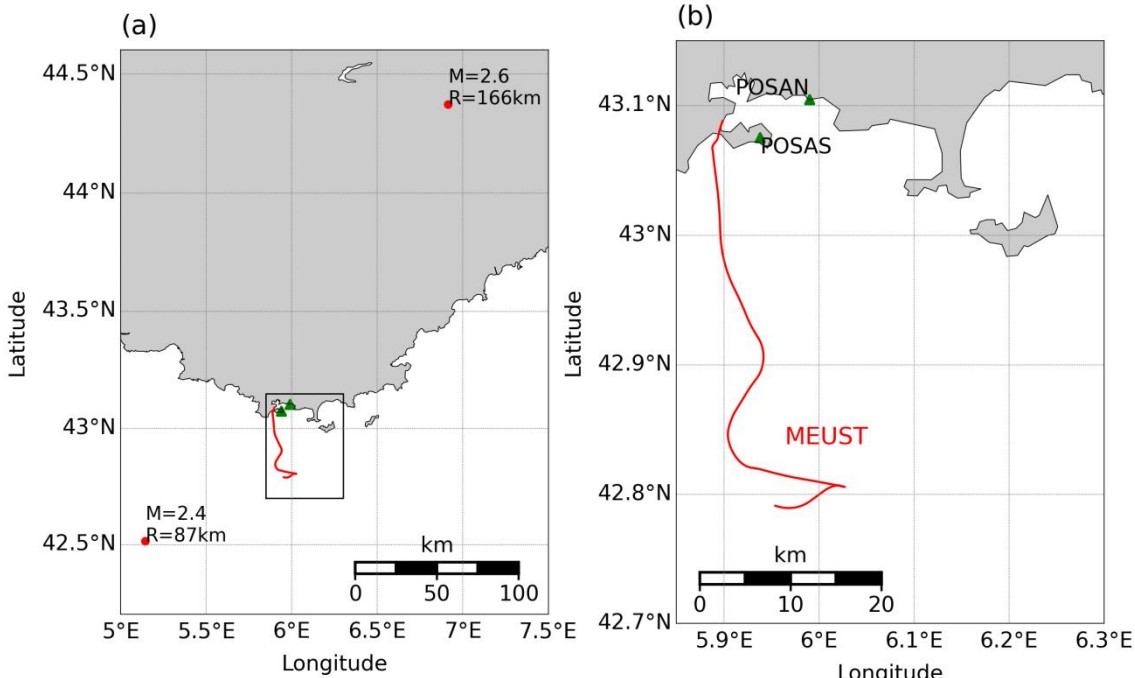

**Figure A2:** Map of earthquake, seismometer and cable locations in Toulon, South of France. The MEUST cable is indicated by a red line, recorded earthquakes are indicated by red circles. Panel (a): catalog magnitudes and hypocentral distances are indicted near earthquake locations. Panel (b) shows the region marked by a black rectangle in panel (a), with cable layout and broadband seismometers (green triangles).

**Appendix B**

Lior and Ziv (2017, 2018) and Luco (1985) derived a set of ground motion RMS descriptions based on Eq. 3 (Eq. 8 in Lior and Ziv, 2018):

$$D_{rms}^{model} = \Omega_0 \sqrt{\frac{\alpha_0}{\pi^{3/2}\kappa T}} \sqrt{G_{1,3}^{3,1}\left(\begin{array}{c}\frac{1}{2}\\0, \frac{1}{2}, \frac{3}{2}\end{array}\middle| \alpha_0^2\right)}, \tag{B1b}$$

$$V_{rms}^{model} = 2\pi\Omega_0\left(\frac{\alpha_0}{(2T)^{1/3}\pi\kappa}\right)^{3/2}\sqrt{\begin{array}{c}2Ci(2\alpha_0)[2\alpha_0 cos(2\alpha_0) + sin(2\alpha_0)] +\\ [\pi - 2Si(2\alpha_0)][cos(2\alpha_0) - 2\alpha_0 sin(2\alpha_0)]\end{array}}, \tag{B1b}$$

and

$$A_{rms}^{model} = (2\pi)^2\Omega_0\left(\frac{\alpha_0}{(2T)^{1/4}(\pi\kappa)^{5/4}}\right)^2\sqrt{\begin{array}{c}2 - 2\alpha_0 Ci(2\alpha_0)[2\alpha_0 cos(2\alpha_0) + 3sin(2\alpha_0)] +\\ \alpha_0[\pi - 2Si(2\alpha_0)][2\alpha_0 sin(2\alpha_0) - 3cos(2\alpha_0)]\end{array}}, \tag{B1c}$$

where $D_{rms}^{model}$, $V_{rms}^{model}$ and $A_{rms}^{model}$ are displacements, velocities and accelerations RMS, respectively, $G_{p,q}^{m,n}$, $Ci$ and $Si$ are the Meijer G, cosine integral and since integral functions, respectively, $\alpha_0 = \pi\kappa f_0$ and $T$ is the used data interval, which is manually chosen in this application. Equations (A1) constitute a set of 3 independent equations with 3 unknowns, for which the observations are obtained in the time-domain ($D_{rms}^{obs}$, $V_{rms}^{obs}$ and $A_{rms}^{obs}$) and the unknowns are the model's spectral

parameters ($\Omega_0$, $f_0$ and $\kappa$). The objective function used is (Eq. 17 in Lior and Ziv, 2018):

$$OF = 100\ max\left(\frac{\left|D_{rms}^{obs+}-D_{rms}^{model}\right|}{D_{rms}^{obs+}},\ \frac{\left|V_{rms}^{obs}-V_{rms}^{model}\right|}{V_{rms}^{obs}},\ \frac{\left|A_{rms}^{obs}-A_{rms}^{model}\right|}{A_{rms}^{obs}}\right), \qquad \text{(B2)}$$

where $D_{rms}^{obs+}$ is the corrected observed displacement RMS: since observed $D_{rms}^{obs}$ is sensitive to low frequencies, it is typically underestimated owing to the signals' limited frequency content. Observed displacement RMS is corrected for the missing frequency content (Eq. 13 in Lior and Ziv,

2018): $D_{rms}^{obs+} = \sqrt{\left(D_{rms}^{obs}\right)^2 + \left(D_{rms}^{corr}\right)^2}$, where $D_{rms}^{corr} = \Omega_0\sqrt{f_I/T}$ (Eq. 15 in Lior and Ziv, 2018). In the

latter, $f_I$ is the lowest resolvable frequency. This approach is implemented on both seismometer recorded, and DAS recorded earthquakes. For seismometers, observed RMS are measured for the vector length of the 3 components, while for DAS, seismogram specific RMS are calculated and averaged for each fiber segment. The best fitting spectral parameters are obtained via grid-search algorithm for $f_0$ and $\kappa$, and $\Omega_0$ is determined for each $f_0$-$\kappa$ combination by a random walk

algorithm. Seismic moments and stress drops are then obtained using Eq. (4). When using Eq. (4a) for DAS recorded S-waves, $\Omega_0$ is multiplied by $\sqrt{2}$ to compensate for the missing horizontal component.

*Code and data availability*: Simulated and observed DAS earthquakes are available on https://osf.io/98cnk/ and https://osf.io/4bjph/, respectively. Broadband seismometer data were acquired by Géoazur: data for the POSAN and POSAS stations were downloaded from RESIF (http://seismology.resif.fr/, last accessed May 2020).


*Author contributions*: IL designed the presented algorithms, performed the analysis and wrote the initial draft; DM designed and performed the simulations; SS helped to adapt and implement the

slant-stack approach to DAS data. AS, JPA, DM, DR and SS contributed to the discussion, methodology, interpretation and presentation of the results.


*Competing interests*: The authors declare no competing interests.

*Acknowledgments*: This work and IL were supported by the SEAFOOD project, funded in part by grant ANR-17-CE04-0007 of the French Agence Nationale de la Recherche. Part of the project was also supported by Université Côte d'Azur IDEX program UCA[JEDI] ANR-15-IDEX-0001 and the Doeblin Federation (FR2800 CNRS). We thank the team from the Centre de Physique des Particules de Marseille who facilitated the access to the MEUST infrastructure. The MEUST infrastructure is
financed with the support of the CNRS/IN2P3, the Region Sud, France (CPER the State (DRRT), and the Europe (FEDER). We thank Stavroula Tsagkli, Katerina Tzamarioudaki and Christos Markou from NCSR Demokritos, the Greek Institute of Nuclear and Particle Physics, who maintain the NESTOR cable infrastructure and facilitated the acquisition campaign. We thank Paris Pagonis from the Hellenic Centre for Marine Research who aided in the access to the HCMR cable. This
infrastructure is part of the European Multidisciplinary Seafloor and water column Observatory (EMSO).

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
