# Peer review of "Strain to Ground Motion Conversion of DAS Data for Earthquake Magnitude and Stress Drop Determination"

_Solid Earth, 2020_

## Author Response (AR1)

We would like to thank both reviewers for their work, and for the concise and constructive comments on the manuscript. The modifications along with replies to all comments are detailed below.

Reviewer #1

(Q1) First, I have to admit that I found the semblance part slightly confusing. Traditionally, semblance is applied on seismic traces – physical measurements with real values. You use an extension to the complex domain, but it is quite unclear because, in Equation 2, h(t) is first defined as the complex portion of the signal and then as its Hilbert transform. If I understood correctly, you are using the signal envelope (i.e., the absolute value of the Hilbert, hence a real argument), so I do not see why the extended semblance is needed.

(R1) We are not working with the envelope but with the analytical signal, $s_{analytic}(t) = s(t) + i\mathcal{H}[s(t)]$, (whose amplitude is the envelope of the signal, $abs(s_{analytic}(t))$). and The real and imaginary parts are real-valued functions, related to each other through the Hilbert transform. Calculating the semblance on the analytical signal is mathematically equivalent to calculating the slant stack of the original signal (first term in the nominator of equation 2) and on its hilbert transform (second term in the nominator of equation 2), and normalizing by the analytic signal's amplitude, i.e., the envelope (denominator of equation 2). We do this to allow for semblance calculation at zero crossings, where calculating the semblance on the signal alone, will yield meaningless results. To clarify that we are working with the analytical signal rather than the envelope, and emphasize the advantage of the analytic signal in zero-crossings, we rephrased the text following equation 2:

"where $2L + 1$ is the number of adjacent stations over which slowness is estimated, $x_j - x_0$ is the distance between station $j$ and the middle station, and $g(t)$ and $h(t)$ are the real and imaginary parts of the analytical signal associated with the seismic trace. The former is the original trace and the latter is its Hilbert transform. The slowness with the highest semblance represents that of the most locally coherent plane wave at the specific time $t$. Including the Hilbert transform of the signal, $h(t)$, amounts to work with the analytical signal, i.e., $g(t) + ih(t)$. In fact, Eq. (2) results from applying the conventional definition of semblance to the analytical signal (Taner et al., 1979). This approach has the key advantage to allow for reliable semblance calculation at the zero-crossings of the original signal, owing to the property that the amplitude of the analytical signal (which is the signal envelope) does not have zero-crossings.".

(Q2) Besides, the discussion about the choice of estimation window is very important, but I think you missed the temporal element. Semblance, at least in the context of seismic exploration, is usually applied over a temporal window. While sample-by-sample semblance is relatively noisy, SNR and resolution can be improved by computing the windowed semblance over a time period that matches the seismic wavelet (or halfwavelet, if you are using the signal instead of its envelope). Finally, I wonder if applying a semblance threshold would be useful for cases in which a certain phase is invisible due to the DAS directivity, such as the P-wave in your synthetic example.

(R2) We agree that applying the semblance over a temporal window will increase the SNR, but it will also introduce a bias in the resolved velocity, especially when waves change their direction (as in the synthetic example) or when velocities vary abruptly. In addition, our use of the analytic signal yields a more reliable sample-by-sample semblance calculation compared with only using the signal. Nevertheless, following the semblance calculation, we smooth the resolved velocities by applying a moving average and accounting for wave direction variations, thus, we are also looking at slowness values averaged according to the signals' wavelet period ($1/f_{min}$).

Regarding a semblance threshold, we attempted to set a threshold in order to clearly identify direct arrivals yet were unsuccessful: analyzed earthquakes are characterized by low SNR (P-waves were not recorded for most observed earthquakes), and for observed earthquakes, the semblance values are very similar for all recorded waves: direct waves, scattered waves, and coherent noise (especially ocean bottom Scholte-waves), as seen in figure 6.

(Q3) I suspect that dispersion effects should be considered during the phase velocity estimation beyond the discussion's few sentences. You filter the signal to the frequency band of interest and estimate a single phase velocity for all frequencies in that band. It is clearly worth verifying if applying the same slant-stack procedure to signals filtered at different frequency bands will yield different phase velocities, at least for the Scholte waves in the synthetic part. If so, you may be able to reconstruct ground motion more accurately by scaling each frequency component with a different phase velocity and hopefully reduce the discrepancy in the later parts of your signal. Park (1999) has a great example of this procedure for Rayleigh waves, but the idea is the same (Multichannel analysis of surface waves, Geophysics).

(R3) We tested the approach using narrow frequency bands for channel #220 of the simulated data as shown in the figure below, added as figure 5 to the revised manuscript. This analysis is done without simulated noise. The top panel shows the lowpass filtered data, equivalent to figure 3 in the manuscript, and subsequent panels show different bandpass filters (frequencies are indicated at the top left corner of the left panels). We repeated the conversion procedure on the bandpass filtered acceleration and strain-rate signals. When comparing the filtered converted strain-rate to filtered ground accelerations, standard deviations are only slightly reduced (values indicated on the top right of the right panels). This suggests that resolving frequency specific phase velocities will not result in a substantial improvement to conversion quality. In addition, using a frequency-specific, along with the already used time-specific, slowness for conversion will make this process extremely complicated. The signal will have to be decomposed into different frequency bands, converted to accelerations, and then recomposed to obtain a converted broad-band signal. Such an approach may require intricate processing, introduce artifacts, and is highly subjective and challenging to implement. Thus, a bandpass limited conversion is not developed and implemented in this manuscript.

The added paragraph and figure are below:

"To examine the performance of the semblance based strain-rate conversion for dispersive waves (e.g., Scholte waves), this technique is applied to strain-rate records filtered at different frequency bands. For this test, noise was not added to the simulated data. An example for the same trace shown in Fig. 3 (#220) is shown in Fig. 5, where 6 different frequency bands are tested: filtered strain-rate data is shown in panels a and a comparison between converted strain-rate (red curves) and accelerations (blue curves) is shown in panels b. Different phase velocities are measured for dispersive waves (2.5-7 seconds) at different frequency bands, while the same velocity is measured for body waves (1-2.5 seconds) (panel c). The agreement between accelerations and converted strain-rate in narrow frequency bands is only slightly better than that obtained for broadband signals, lowpass filtered at 12 Hz (top line in panels a and b), as indicated by the standard deviations of the residuals (top right corners of panels b). The small difference between bandpass filtered (e.g., 4.8-7.6Hz in Fig. 5) and broad band (0-12Hz in Fig. 5) standard deviations of the residuals suggests that the dispersive nature of these waves has a small effect on conversion quality. Thus, applying time-specific and frequency-specific slowness to convert broadband seismic signals is unlikely to result in a significant improvement in conversion robustness. Also, such an approach may require intricate processing, introduce artifacts, and is highly subjective and challenging to implement. Thus, this analysis is focused on the more generic case of a broadband signal, and a bandpass limited conversion is not developed and implemented.".

[Figure]

(Q4) I also think that the comparison to the alternative approach is slightly unfair. Scaling with a constant slowness is an unrealistically bad option, especially for the field data – a phase velocity of 400 m/s for body waves! The naive but more reasonable alternative would be to apply this conversion in the FK domain. Since it is straightforward to implement, I recommend using it as a baseline for your comparisons. You will certainly need relatively large windows, but it is still much better than a constant slowness. I first emphasize this point because, strictly speaking, you do not need to estimate the different phases' velocities except as a scaling factor. Therefore, the delicate interpretation that you correctly mention in the introduction can be skipped. Second, even a constant slowness is not terrible from a source inversion standpoint, so I wonder how the FK alternative approach will perform.

(R4) We replaced the baseline comparison to an FK based approach. Changes were thus made throughout the manuscript. The FK approach generally yields lower velocities owing to its poorer slowness resolution, especially for high velocities. We can reliably conclude that the semblance derived slowness values are the more reliable ones, as seen by the

synthetic analysis. The FK alternative does yield similar source parameters as seen by our added figure 10 (replacing the objective function comparison), however, since FK slowness are generally higher than semblance slowness, FK magnitudes are lower than semblance magnitudes. Since FK magnitudes are biased, a comparison between FK derived source parameters and catalog and seismometer estimates are not shown.

Many small changes were made throughout the manuscript owing to this change. The major changes are detailed here:

1) To the end of section 2 we added a paragraph detailing the FK approach: "Throughout this manuscript, the semblance based slowness determination method is compared to an FK based method, applied as follows. Frequency-wavenumber transforms are calculated on the filtered strain (rate) signals in consecutive windows using the same number of adjacent traces used for semblance calculation. High amplitude temporal frequency ($f$) - spatial frequency ($v$) combinations are identified as those whose spectral amplitudes are higher than the 99th percentile of all amplitudes in the FK domain. The spectral amplitudes are summed separately for the two FK quadrant (positive $f$ and positive $v$ or positive $f$ and negative $v$) and slowness is estimated using data from the higher sum quadrant by fitting $f = v/p_x$. The slowness time-series is then smoothed and used for strain (rate) to ground motion conversion in the same manner previously described for the semblance analysis."

2) several sentences were added to sections 3.2 and 5 interpreting the results of the FK approach and its comparison to the semblance approach.

The paragraph and figure discussing the difference in objective function for source parameter inversion using the 2 approaches was changed to a discussion of magnitude difference (section 6.2): "To further compare the performance of FK and semblance based slowness, magnitude estimates are compared. Figure 10 plots the difference between magnitudes estimated following the FK and semblance based conversion schemes as a function of catalog magnitude. Magnitudes estimated for data converted using FK derived slowness are generally lower than those resolved using semblance derived slowness. This trend is expected given the lower apparent velocities determined via the FK approach, as previously shown for both simulated (Fig. 3) and observed (Fig. 7 and 8) earthquakes.".

3) Figure 10 in the original manuscript showing the different spectra of the 2 conversion approaches was removed since spectra are similar and produce similar source parameter objective functions. As expected FK converted acceleration spectra present lower amplitudes than semblance converted acceleration spectra.

4) A paragraph was added to the beginning of the discussion section, discussing the difference between the 2 approaches and the advantages of the semblance based approach: "The comparison between semblance

derived slowness and FK derived slowness for strain (rate) to ground motions conversion reveals several advantages favoring the semblance based approach. Unlike semblance analysis, whose implementation and interpretation is simple and objective, FK analysis introduces considerable subjectivity into the slowness determination procedure since interpreting an FK image may be done in various ways. The FK analysis generally yielded lower velocities owing to its poor slowness resolution, as observed for both simulated and recorded earthquakes. For these reasons, when comparing simulated accelerations and converted strain-rates, a significantly better agreement was obtained using the semblance based approach. The ground motion conversion difference between FK- and semblance-based approaches is most pronounced for direct arrivals, i.e. P- and S-waves. Thus, the semblance based conversion is particularly advantageous for EEW, when short duration, relatively fast propagating waves, are required for speedy and reliable source parameter estimation.".

(Q5) The paper also does not address fiber coupling issues, which are the main drawback in using DAS data acquired from existing cables. I wonder if the first arrivals from known, distant sources, can be used to "track" fiber coupling, as their changes along the array should be quite gradual. It is also worth mentioning, in my opinion, that your scaling accuracy depends on the geometry of the problem. Not only some phases will be weaker because of DAS directivity, but their scaling will also be noisier (as happens with the synthetic P-wave). Along that line, it would be useful to see a map of the earthquakes you used and not only a distance-depth plot, along with some coarse explanation of the apparent velocities you observe.

(R5) We used ocean-bottom fibers, that were not intended for seismology, as such many fiber segments displayed signals that were not sufficiently uniforms for the procedures described in the manuscript. We manually selected several sections on each fiber for which the coupling was sufficiently uniform. We added a paragraph and figure to the conclusions showing that coupling variations within the selected segments are not substantial. We did so by examined the variability in average absolute amplitude (found to be more stable than direct arrivals) along these segments. The added a figure (figure 14) and paragraph:

"A significant hinder when using DAS is the mechanical coupling between the fiber and the solid Earth. This issue is particularly troublesome when standard telecommunication fibers are used, specifically those deployed underwater, as their coupling quality is often unknown and may prevent reliable seismic monitoring (e.g., Sladen et al., 2019; Lior et al., 2021). Here, specific segments for which recording quality is sufficiently uniform were manually identified. The effect of coupling along these limited segments is quantified by considering the signals' average absolute strain-rate

amplitudes in decibels (dB), plotted in Fig. 14 for all traces, segments and cables. In Fig. 14, only earthquakes recorded at hypocentral distances longer than 80 km are plotted, to ensure slow propagation related amplitude changes along the fiber. The variabilities are generally small, limited to 2-3 dB with several exceptions of ~8 dB. These mostly minor deviations, along with the small DAS magnitude uncertainties (vertical errorbars in Fig. 11 indicate standard deviations of magnitudes resolved independently for each DAS channel) indicate that these segments display sufficiently uniform coupling for ground motion conversion and source parameter estimation. Moreover, the fact that even for non-uniform coupled cables as those used here, sufficiently uniform coupling is observed, even if in limited segments, demonstrates the potential of underwater fibers for reliable source parameter estimation.".

[Figure]

We mention the effect of the geometry of the problem in the 3rd paragraph in the discussion, referring to the lower than expected velocities resolved for synthetic P-waves: "…Careful filtering of DAS signals needs to be applied to isolate specific plane waves from DAS earthquake records, as demonstrated in part in Fig. 5, yet conversion errors may still result from inadequate slowness resolution, incoherent plane waves, and noise. The effect of noise is seen in Sect. 3 where synthetic P-waves suffer from low SNR, which results in lower than expected apparent velocities, and a slightly reduced conversion quality. The amplitudes of DAS P-waves, which arrive at near-vertical incidence angles with respect to horizontal fibers, are especially low since they are reduced by a factor of squared cosine of incidence angle (e.g. Mateeva et al., 2014). Further consideration of these issues is beyond the scope of this manuscript. In-spite of these complexities,

converted DAS signals allow for reliable magnitude estimation, demonstrating the robustness of the conversion procedure.".

To the end of the first paragraph in section 4 we added the following: "The earthquake DAS records used in this study are dominated by scattered Scholte-waves, with velocities between 100 and 400 m s$^{-1}$ (Lior et al., 2021).".

We added maps as Appendix A. Appendix A in the previous version was changed to Appendix B.

(Q6) Finally, a visual comment - in many figures (2-d, 3-b, 8-d, 10), some elements are practically invisible in the legend and hard to see in the plot. Please edit them. In Figure 9, I couldn't understand which symbols represent a constant slowness.

(R6) We modified the mentioned figures. Figure 9 (currently figure 10 of the revised manuscript) has been changed. The y-axis values are the difference between the magnitudes using FK slowness and semblance slowness. The first line of the figure caption was changed to: "Magnitude discrepancies: Magnitudes estimated following the FK based slowness conversion minus magnitudes estimated following the semblance based conversion, plotted as a function of catalog magnitude.".

(Q1) When working with DAS, one thing that comes up again and again is the problematic of instrument coupling and response. In this manuscript, there is no real discussion about this, and the DAS data is assumed to be true ground strain (rate). I feel like there could be either an analysis how the coupling / response could affect the source parameter estimation in the end, or a quantitative sensitivity analysis.

(R1) Please see (R5) to reviewer #1.

(Q2) Page 2, Line 46-47: Whereas I agree that converting strain to displacement is a possible approach to be able to determine source parameters, I am wondering if it were possible "the other way around" as well. The mentioned source models all date back to the 70s, and I am curious if source models based on strain observations would be another way to approach this problem. I am not an expert on source parameter estimation, but I think there may be some benefits in re-visiting existing theory. I am wondering if you attempted/considered that? The key here being "the ability to invert for the source properties using conventional methods". What about new methods, developed with a focus on strain as observation?

(R2) We are not aware of any attempts or formulations to directly resolve source parameters using strain records, and believe that this cannot be achieved because of the sensitivity of strain records to local properties and heterogeneities of the media.

Implemented source models relate ground motions to earthquake source parameters, as well as propagation and media properties. These ground motions are directly related to the source time function using a Green's function (e.g., equation 4.32 of Aki and Richards, 2002). In contrast to ground motions, which are directly related to source slip, strain measurements are greatly affected by local media properties and heterogeneities (e.g., van den Ende and Ampuero, 2020; Singh et al., 2020). Thus, the same recorded strain amplitudes may result from high amplitude ground velocities propagating at high apparent velocities or low amplitude ground velocities propagating at low apparent velocities (equation 1). Converting strain records to ground motions effectively accounts for the effect of local media and allows for direct evaluation of source parameters. As discussed in the text, in addition to using the apparent velocity, ground motions may be obtained via integration in space (e.g., van den Ende and Ampuero, 2020) or approximated using the gauge length (Lellouch et al., 2020).

(Q3) Page 3, Line 87 and Page 3, line 90: This seems to be eq. 2 from Shi and Huo (2019), where your f corresponds to the seismic trace (they call that "real part") and h to the Hilbert transform (you call that "imaginary part"). Maybe clarify this, I found the formulation confusing, because you mention both "imaginary part" and "Hilbert transform". Otherwise, it sounds to me like that the seismic trace f(t) (which is a time domain signal of real-valued numbers) has an imaginary part h.

The process of this complex valued trace is explained nicely in the paper you cite, and I think it would help the general understanding if you could explain this here a bit more clearly.

(R3) We revised the text surrounding equation 2. Please see (R1) to Reviewer #1.

(Q4) Page 4, Line 107: The formulation about the combination of signals, including potentially dispersive waves, is a bit vague. If you filter within a frequency band of interest, there still might be dispersive waves within this frequency window, and there still may be different arrivals on the array simultaneously.

(R4) We added the following after this line to elaborate on the expected filtered signal: "Filtering will reduce noise and limit dispersive effect, however, simultaneous wave arrivals, complex propagation and dispersion effects are still expected".

Also see (R3) to reviewer #1.

(Q5) Page 4, Line 110: If I read this correctly, you allow the phase velocity to be within +/- 100 m/s, with spacings of 5 km/s. Is this correct or is there a "k" missing (for km/s) in the outer bounds of the allowed slowness (inverse of apparent phase velocity).

(R5) We test different slowness values between -0.01 s/m and 0.01 s/m (passing through zero) with spacings of 0.0002 s/m, i.e. 100 equally spaced slowness values. We revised this line: "The range of examined slowness values is chosen to be between -0.01 s/m and 0.01 s/m with 0.0002 s/m slowness intervals (i.e., 100 slowness values), and at each time…".

(Q6) Page 4, Line 115: If you calculate the slowness including the sign, you mention that you get the direction back – so why do you need to go through the process of a moving average? It would be great to explain this process in more detail, and why it is required.

(R6) We added explanations for the abrupt variations of semblance value and sign, and the averaging of the absolute slowness values. The revised section is as follows: "The slowness time series (derived from semblance) may often be characterized by abrupt variations of value and sign (i.e., propagation direction) **owing to complex wave propagation, interference and dispersion effects**. Thus, a moving average, with window size set to be equal to the signal's longest period of interest, is applied to the absolute value of the slowness **(preventing the averaging of slowness values with different signs)**. The sign is then determined as the one that dominates each averaged window, i.e., the most recurrent sign. absolute value of the slowness. The sign is then determined as the one that dominates each averaged window, i.e., the most recurrent sign.".

Also see (R2) to reviewer #1.

(Q7) Page 4, Line 118: When talking about filtering, it would be great to specify which filter you use. I assume you are talking about a Butterworth bandpass filter here? (After further reading, indeed you mention this specifically later in the text).

(R7) At the end of this section we added the following: "In following sections the same 4-pole zero-phase Butterworth filter is used for both filtering operations.".

We removed the filter description from the beginning of section 5.

(Q8) Page 7, Line 148: Great idea to add real ambient noise from recordings to these simulations. What is the reasoning behind not simulating noise in this frequency band? Are the added noise waveform recordings added to each channel independently, with the same spatial resolution as the numerical simulation? It would be great to get some more background on this added noise. Also, the depth of the noise recordings is from 800 meters, whereas the depth of the water column in the numerical example is 20 m. Do you expect this to have any effects on the actual measurements? What was the gauge length of the recorded noise waveforms? How did you "spatially differentiate"? Such that you did include the gauge-length effect? I think a little bit more detail would be beneficent for the readers here.

(R8) We differentiated the simulated velocity records in space to obtain strain-rates. Our differentiation assumes that we are analyzing long wavelengths compared to the gauge length, so it is simply: $\dot{e}(x) = \frac{V(x+GL/2)-V(x-GL/2)}{GL}$. We felt that adding recorded DAS noise best represents both the structure of instrumental noise and correlated underwater ambient noise (secondary microseisms), for a more reliable comparison to recorded earthquakes in following sections. DAS ambient noise was recorded using a gauge length and spatial sampling of 10 m and sampling rate of 100 Hz, while data was simulated using a gauge length of 10 m, spatial sampling of 5 m and sampling rate of 200 Hz. Thus, ambient noise was interpolated in both space and time to 5 m and 200 Hz. Interpolated ambient noise and simulated DAS data were then added, keeping the spatial structure of both. We used this noise simply to simulate correlated noise in a similar frequency content to that of the signal, and we don't expect this noise to have any water depth related effect on the measurements.

This paragraph was revised to: "Simulated velocity waveforms were differentiated in time and space to obtain ground accelerations and strain-rates, respectively, noise was added to the later, and the ability to convert the latter to the former via the proposed slant-stack approach is examined. Strain-rate records were calculated as $\dot{e}(x) = \frac{V(x+GL/2)-V(x-GL/2)}{GL}$ and ground acceleration records were calculates as $A(t) = \frac{V(t+dt)-V(t-dt)}{2dt}$, where V, $GL$ and $dt$ are the simulated velocity, gauge length and temporal sampling, respectively. Simulated strain-rate signals are thus characterized by gauge length and spatial sampling of 10 m and 5 m, respectively. To reliably generate ocean-bottom DAS records, ambient noise measurements were added to simulated strain-rates, keeping the noise records' spatial correlation (Fig. 2c). In the simulated water depth (i.e., 20 m), ocean-bottom DAS records are typically dominated by surface gravity waves (e.g., Lior et al., 2021), which may be easily filtered due to their lower frequency content compared to the simulated earthquake. Thus, ambient noise recorded at a water depth of ~800 meters was used. These records are composed of instrumental noise and secondary microseisms in similar frequencies to those of the simulated earthquake (Fig. 2d). The added noise measurements were recorded on 22 July 2019 by an underwater cable deployed offshore Toulon, South of France (Sect. 4). Noise records, sampled at 100 Hz and 10 m (spatial sampling is equal to the gauge length), were resampled to match the simulated data using a 2D interpolation function. Noise records were then differentiated to strain-rate and rescaled to simulate challenging noise conditions, with an average signal-to-noise ratio (SNR) of 8.2 (Fig. 2d). Here, SNR was calculated as the root-mean-squares (RMS) ratio of average signal and noise amplitude spectra between 2 and 12 Hz. In-spite of the added noise, accelerations converted from strain-rates are compared to simulated accelerations

(derived from simulated velocities by finite-difference time-derivative). Noise was not added to the latter, constituting a stringent algorithm validation.".

(Q9) Page 7, Line 160: Are the mentioned wavelengths the apparent wavelengths along the fiber (propagation direction considered), or are they simply calculated by 12 Hz/Velocity? The apparent velocity along the fiber for not in-line events could potentially be much higher than this value?

(R9) It is the apparent P-wave wavelength, measured along the vertical axis in figure 2b. The word "apparent" was added.

(Q10) Page 8, Line 165: Is this the apparent slowness?

(R10) Yes, the word "apparent" was added.

(Q11) Page 8, Figure 3 caption: The last sentence does read a bit odd. Maybe change to "The signals in the gray regions have been amplified by a factor of 6 for easy comparison…."

(R11) We agree. The last sentence was revised as suggested.

(Q12) Page 9, Line 186 and throughout this entire section: The example you provide is very impressive. Yet I feel like it is an unfair comparison, since you use the same apparent phase velocity for the conversion (dotted line) for all arrivals. Estimating these for separate windows would potentially also give you a better estimate of the strain-rate derived acceleration. You also mention the Frequency-Wavenumber (FK) domain approach in your introduction, but do not compare the slant-stack conversion to this approach. One may argue that for your given example, the FK domain approach would result in a very similar converted acceleration that you get with the slant-stack conversion. So, it feels a bit "unfair" to compare your method to basically the simplest existing method with even violated assumptions (not a single plane-wave arrival). Because your method seems to work quite well, I think it would be great to compare it to the FK-domain approach.

(R12) We replaced the comparison of constant velocity to a comparison with slowness derived from FK calculations. Please see (R4) to reviewer #1.

(Q13) Page 16, L 287: Is this parameter tuning choosing parameters based on existing literature? Or is this "tuning" actually an iterative process until you arrive at these values?

(R13) The following parameters are model based: $U_{\varphi\theta}$, $F_s$ and $k$. They are derived from commonly used models which are referenced in the text (a missing reference to Madariaga, 1976 was added). In contrast, $\rho$, $C_P$ and $C_S$ are not model based and are assigned with average values, previously used by other studies (e.g., Lior and Ziv, 2020). Since these parameters represent the source regions, they are used for both ocean-bottom and on-land observations. A reference to Lior and Ziv (2020) was also added.

(Q14) Page 16, line 289: Cs should be C_s?

(R14) Yes, it has been corrected.

(Q15) Page 20, Figure 11: Are the DAS-derived magnitudes here the ones from the slant-stack conversion approach? It would be good to see the estimated magnitude comparison here for both strain to displacement conversion methods, in order to see how the 'error' in the conversion propagates to the final magnitude estimates. The same holds for Figures 12 and 13. I would understand if you think that Figure 9 is sufficient for this comparison, I just think that it would be nice to see this here as well base on my personal preference on how I look at figures.

(R15) In Figures 11-13 we plot the semblance based source parameters. Following the change in reference conversion method and the comparison to the FK slowness method, several figures changed (as detailed in (R4) to reviewer #1). We changed Figure 10 (previously 9) to a plot comparing magnitude estimates using semblance based conversion and FK based conversion. This plot shows that FK based magnitudes are generally lower than semblance based magnitudes by ~0.1 magnitude units. Since this plot shows the magnitude differences between the 2 different conversion methods very well, we did not add a comparison to FK magnitudes in Figure 11.

A comparison between the agreement of FK and semblance magnitudes and seismometer magnitudes will be misleading since we do not seek the best agreement between DAS and seismometer magnitudes, rather the most reliable ground motion conversion. Since the DAS fiber and the seismometers are not co-located, we do not expect their source parameters to be in full agreement. The disparity between the two can be due to many factors in addition to ground motion conversion reliability: different site response, different propagation characteristics (DAS and seismometers are not co-located as seen in the maps in Appendix A), different source to station back-azimuth and radiation pattern, and more. Thus, a comparison between the source parameters of FK DAS and seismometers is not shown.

To the paragraph discussing figures 11-13 (section 6.2) we added the following sentence: "Since the DAS fiber and the seismometers are not colocated, their estimated source parameters are not expected to be in perfect agreement.".

(Q16) Page 22, Line 375: It would be great to quantify "in good agreement" here.

(R16) The agreement between corner frequencies and stress drops is quantified via the standard deviations to parameter residuals, reported in the legends of figure 12. These standard deviations essentially indicate the within-event variability of $f_0$ and $\Delta\tau$ estimates between specific DAS segments and seismometers. Thus, these within-event variabilities may be compared to within-event variabilities reported in figure 8 of Lior and Ziv (2018), calculated using the same source parameter inversion approach. The figure from Lior and Ziv (2018) and figure 12 of the manuscript are given here:

[Figure]

[Figure]

[Figure]

[Figure]

[Figure]

The end of this paragraph was modified: "…Thus, corner frequencies and stress drops are not well constrained. **The standard deviations to parameter residuals (Fig. 12) are within-event variabilities between the estimates of specific DAS segments and seismometers. That these values are only slightly higher than within-event variabilities reported by Lior and Ziv (2018) suggests that** In-spite of the inability to reliably determine these parameters, DAS and seismometer derived $f_0$ and $\Delta\tau$ are found to be in good agreement (Fig. 12).".

(Q17) Page 23, Line 397: When talking about real-time applications utilizing DAS data, the implementation due to the large amounts of data could become an issue. Did you do some back of the envelope calculations on how long such an inversion would take in real-time for the investigated earthquakes? If this would indeed be possible in real-time, I think your method does look very promising.

(R17) The algorithm, as is, is not adapted for real-time applications. Several issues need to be addressed and the procedure will have to be modified. The implemented source parameter inversion is also not adapted for real-time applications owing to its high computational cost. A different, more efficient algorithm will need to be applied (e.g., Lior and Ziv, 2020). All these will be addressed in a subsequent paper. We added the following to this paragraph: "To this end, the proposed algorithm will need to be adapted for real-time performance.".

In the end of the Conclusions section, we state that "The algorithm for strain (rate) conversion may be adapted for real-time applications and used in conjunction with real-time source parameter determination schemes (e.g. Lior and Ziv, 2020) for a DAS-based EEW system".

(Q18) The reference to Singh et al., 2020 seems to be missing in the List of references. Please make sure that all references are there.

(R18) Done.

General remarks to figures:

(Q19) The figures are not inserted over the full width due to an expected two-column layout. I am aware that we live in digital times, and that we can always zoom into our pdfs – but by doing this for some of the figures, I felt the resolution was a little bit too low. I am unsure what the submission policy for figures (do they need to be within a certain file size?) is, but I hope that the final publication has higher resolution figures, such that zooming in actually reveals more information.

I also think that the figure captions and labels should be slightly increased in text size.

Some figures (e.g. Figure 8, panel (e)) might not be easily distinguishable by people affected by colorblindness. Most of the figures use a colormap that seems to be "viridis", which is a great choice. I would change the colormap of Figure 8, panel (e) to this as well (also for Figure 2, panel (a). Generally I think it is important to use colors that can be distinguishable by colorblind people, or use different line-styles whenever possible (e.g. dashed, dotted etc..).

(R19) The resolution was drastically reduced in the downloaded pdf. We have enlarged the figures in the manuscript (and thus the font sizes as well) and hope that the final public version will have higher resolution.

We made other modifications to the figures according to (R6) of reviewer #1 and also changed the color map to viridis.

---

## Author Response (AR2)

Dr. Gilda Currenti
Editor
Solid Earth

08/04/2021

Re: Manuscript reference number se-2020-219.

Dear Dr. Gilda Currenti,

We would like to thank you for your work, and we thank the reviewers for their remarks. The modifications along with replies to all comments are detailed below.

Yours sincerely,

Itzhak Lior.

General modifications:

In the manuscript's title, we propose to replace the acronym "DAS" with "Distributed Acoustic Sensing".

Response to editor comments:

(Q1) Line 88-98: the description of the method is more clear, but there is still a repetition in the definition. Please, rephrase the description, be more direct by saying that the semblance it is applied to the analytical signal and define it once.

(R1) The text following Equation 2 was revised as follows:
"...where $2L + 1$ is the number of adjacent stations over which slowness is estimated, $x_j − x_0$ is the distance between station $j$ and the middle station, $g(t)$ is the seismic trace and $h(t)$ is the Hilbert transform of $g(t)$. The slowness with the highest semblance represents that of the most locally coherent plane wave at the specific time t. Including the Hilbert transform in Eq. (2) is equivalent to applying the conventional definition of semblance (Taner et al., 1979) on the analytical signal $g(t) + ih(t)$. This approach has the key advantage of allowing for reliable semblance calculation at the zero-crossings of the original signal, owing to the property that the amplitude of the analytical signal (which is the signal envelope) does not have zero-crossings.".

(Q2) Line 209-216: All the plots are in slowness. Please, report the results in slowness in order to directly compre with figs. You can add the value in velocities in parenthesis.

(R2) Semblance values are reported while equivalent velocities are given in parenthesis.

(Q3) Caption Figure 4: Check the description of panel (d). Should it be obtained by FK analysis?

(R3) This line was revised: "...converted to ground accelerations: using **semblance-derived** slowness (panel c), and **FK-derived** (panel d)".

(Q4) Line 250: correct the typo "in panels a and a".

(R4) We added a comma for better readability of this sentence. Its first half presents panel a, and the second half (following the new comma) presents panel b: "filtered strain-rate data is shown in panels a**,** and a comparison between converted strain-rate (red curves) and accelerations (blue curves) is shown in panels b".

(Q5) Line 549: change "Equations (A1)" in "Equations (B1)".

(R5) Done.

Response to Reviewer #2:

(Q1) Colorbars in Figures 2,4, and 5 are missing for the waveform plots.

(R1) Color bars were added to figure 2.

The color bar in figure 4 refers to all relevant panels (a and c-f), the following was added to the figure caption: "The color code is uniform for panels (a) and (c)-(f) and indicated in the colorbar in the top row.".

In figure 5, the color code is different for each panel. A uniform color code will not work due to the large differences between the signals. We added the min and max micro-strain-rate values plotted at each panel, to the top right corner. The following was added to the figure caption: "...the minimum and maximum plotted micro-strain-rate values are indicated at the top right of each panel".